# National and subnational short-term forecasting of COVID-19 in Germany and Poland during early 2021

## Abstract

**Background** During the COVID-19 pandemic there has been a strong interest in forecasts of the short-term development of epidemiological indicators to inform decision makers. In this study we evaluate probabilistic real-time predictions of confirmed cases and deaths from COVID-19 in Germany and Poland for the period from January through April 2021.

**Methods** We evaluate probabilistic real-time predictions of confirmed cases and deaths from COVID-19 in Germany and Poland. These were issued by 15 different forecasting models, run by independent research teams. Moreover, we study the performance of combined ensemble forecasts. Evaluation of probabilistic forecasts is based on proper scoring rules, along with interval coverage proportions to assess calibration. The presented work is part of a pre-registered evaluation study.

**Results** We find that many, though not all, models outperform a simple baseline model up to four weeks ahead for the considered targets. Ensemble methods show very good relative performance. The addressed time period is characterized by rather stable non-pharmaceutical interventions in both countries, making short-term predictions more straightforward than in previous periods. However, major trend changes in reported cases, like the rebound in cases due to the rise of the B.1.1.7 (Alpha) variant in March 2021, prove challenging to predict.

**Conclusions** Multi-model approaches can help to improve the performance of epidemiological forecasts. However, while death numbers can be predicted with some success based on current case and hospitalization data, predictability of case numbers remains low beyond quite short time horizons. Additional data sources including sequencing and mobility data, which were not extensively used in the present study, may help to improve performance.

## Plain language summary

We compare forecasts of weekly case and death numbers for COVID-19 in Germany and Poland based on 15 different modelling approaches. These cover the period from January to April 2021 and address numbers of cases and deaths one and two weeks into the future, along with the respective uncertainties. We find that combining different forecasts into one forecast can enable better predictions. However, case numbers over longer periods were challenging to predict. Additional data sources, such as information about different versions of the SARS-CoV-2 virus present in the population, might improve forecasts in the future.

Short-term forecasts of infectious diseases and longer-term scenario projections provide complementary perspectives to inform public health decision-making. Both have received considerable attention during the COVID-19 pandemic and are increasingly embraced by public health agencies. This is illustrated by the US COVID-19 Forecast[1,2] and Scenario Modeling Hubs[3], supported by the US Centers for Disease Control and Prevention, as well as the more recent European COVID-19 Forecast Hub[4], supported by the European Center for Disease Prevention and Control (ECDC). The Forecast Hub concept, building on pre-pandemic collaborative disease forecasting projects like FluSight[5], the DARPA Chikungunya Challenge[6], or the Dengue Forecasting Project[7] aims to provide a broad picture of existing short-term projections in real time, making the agreement or disagreement between different models visible. Also, it forms the basis for a systematic evaluation of performance. This is a prerequisite for model consolidation and improvement, and a need repeatedly expressed[8]. It has been highlighted that such modeling studies should be prospective[9] and ideally follow pre-registered protocols[10] in order to prevent selective reporting and hindsight bias (i.e., the tendency to overstate the predictability of past events in hindsight).

We here report on the second part of a prospective disease forecasting study, pre-registered on 8 October 2020[11] and including forecasts made between 11 January 2021 and 29 March 2021 (with last observed values running through April; twelve weeks of forecasting). It is based on the German and Polish COVID-19 Forecast Hub (https://kitmetricslab.github.io/forecasthub/), which gathers and stores forecasts in real time. This platform was launched in close exchange with the US COVID-19 Forecast Hub in June 2020. In April 2021, it was largely merged into the European COVID-19 Forecast Hub, shortly after the latter had been initiated by ECDC. During our study period, fifteen independent modeling teams provided forecasts of cases and deaths by the appearance in publicly available national-level data, provided either by national health authorities (Robert Koch Institute, RKI[12] or the Polish Ministry of Health, MZ[13]; the primary data source) or the Johns Hopkins University Center for Systems Science and Engineering (JHU CSSE; refs. [14] and[15]). As specified in our study protocol, we report results on forecasts up to a horizon of four weeks, but focus on forecasts one and two weeks ahead. While we acknowledge the relevance of longer horizons for planning purposes, we argue that factors like changing non-pharmaceutical interventions and the emergence of new variants limit meaningful forecasts (as opposed to scenarios) to rather short time horizons, especially for cases. Also, we focus almost exclusively on incident quantities, as their cumulative counterparts have almost completely vanished from any public discussion.

The time series of cases and deaths in both countries are displayed in panels (a) and (b) of Fig. 1. The study period covered in this paper is marked in dark gray, while the light gray area represents the time span addressed in the first part of our study[16]. Our study period contains the transition from the original wild-type variant of the virus to the B.1.1.7 variant (later called Alpha). Panel (c) of Fig. 1 shows the estimated weekly percentages of all cases which were due to the B.1.1.7 variant in Germany[17] and Poland[18,19] in calendar weeks 4–12. Panel (d) shows the proportion of all performed PCR tests which turned out positive. While in Germany, the curve follows a U-shape similar to the case incidence curve, the test positivity rate continuously increased in Poland, peaking at 33%. Panel (e) shows the Oxford Coronavirus Government Response Tracker (OxCGRT) Stringency Index[20]. It can be seen that compared to the first part of our study, the level of non-pharmaceutical interventions was rather stable at a high level during the second period. We note, however, that on 27 March, a new set of restrictions was added in Poland

(closure of daycare centers, hair salons, and sports facilities, among others), which is not reflected very strongly in the stringency index. The start of vaccination rollout in both countries coincides with the start of our study period. However, by its end, only roughly one-sixth of the population of both countries had received the first dose, and roughly one-twentieth had received two doses (with the role of the one-dose Johnson and Johnson vaccine negligible in both countries); see panel (f). Note that all these data are publicly available via the respective public health agencies and their use does not require ethical approval.

We find that averaged over the second evaluation period, most though not all of the compared models were able to outperform a naïve baseline model. Heterogeneity between forecasts from different models was considerable. Ensemble forecasts combining different available predictions achieved very good performance relative to single-model forecasts. However, most models, including the ensemble, did not anticipate changes in trend well, in particular for cases. Pooling results over both evaluation periods, we find that ensemble forecasts for deaths were well-calibrated (i.e., prediction intervals contained the true value roughly as often as intended) even at longer prediction horizons and clearly outperformed baseline and individual models, while for cases, this was only the case for one- and to a lesser degree two-week-ahead forecasts.

## Methods

The methods described in the following are largely identical to those in the first part[16] of our study, but are presented to ensure self-containedness of the present work.

**Targets and submission system.** Teams submitted forecasts for the weekly incident and cumulative confirmed cases and deaths from COVID-19 via a dedicated public GitHub repository (https://github.com/KITmetricslab/covid19-forecast-hub-de). For certain teams running public dashboards, software scripts were put in place to transfer forecasts to the Forecast Hub repository. Weeks were defined to run from Sunday through Saturday. Each week, teams were asked to submit forecasts using data available up to Monday, with submission possible until Tuesday 3 p.m. Berlin/Warsaw time (the first two daily observations were thus already available at the time of forecasting). Forecasts could either refer to the time series provided by JHU CSSE or those from Robert Koch Institute and the Polish Ministry of Health. All data streams were aggregated by the time of the appearance in national data, see also Supplementary Note 4 of Bracher et al.[16]. Submissions consisted of a point forecast and 23 predictive quantiles (1%, 2.5%, 5%, 10%, …, 95%, 97.5%, 0.99) for the incident and cumulative weekly quantities. As in previous work[16], we focus on the targets on the incidence scale. These are easier to compare across the different data sources than cumulative numbers, which sometimes show systematic shifts.

**Evaluation metrics.** As forecasts were reported in the form of 11 nested central prediction intervals (plus the predictive median), a natural choice for evaluation is the interval score[21]. For a central prediction interval $[l, u]$ at the level $(1 - \alpha)$, thus reaching from the $\alpha/2$ to the $1 - \alpha/2$ quantile, it is defined as

$$\text{IS}_\alpha(F, y) = (u - l) + \frac{2}{\alpha} \times (l - y) \times \chi(y < l) + \frac{2}{\alpha} \times (y - u) \times \chi(y > u),$$

(1)

where $\chi$ is the indicator function and $y$ is the realized value. Here, the first term characterizes the spread of the forecast distribution, the second penalizes overprediction (observations fall below the prediction interval) and the third term penalizes underprediction.

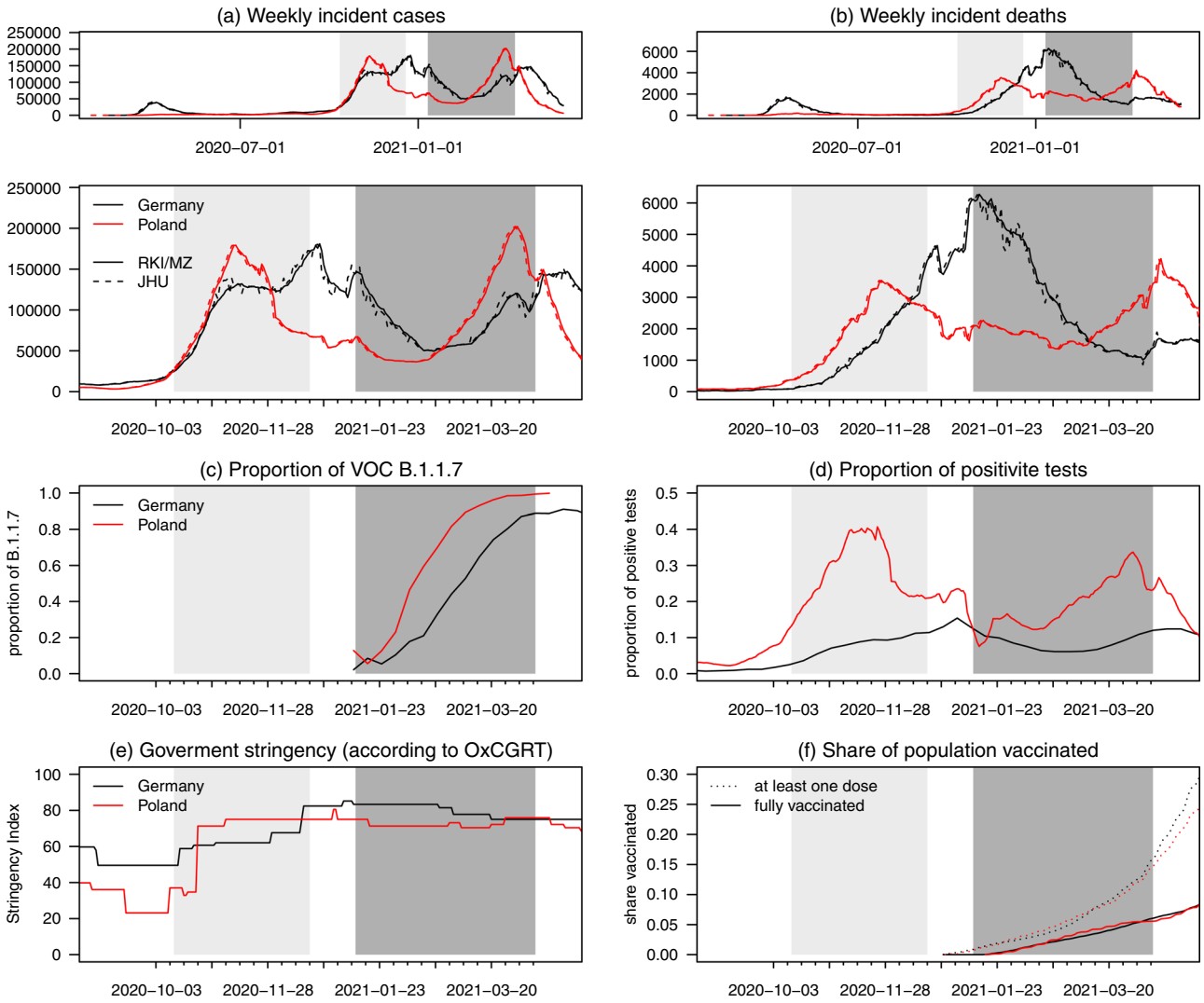

**Fig. 1 Overview of relevant epidemiological time series.** Reported cases (**a**) and deaths (**b**) per seven days in Germany (black) and Poland (red) according to Robert Koch Institute, the Polish Ministry of Health (MZ; solid lines), and Johns Hopkins CSSE (dashed). Additional panels show **c** the share of cases due to the B.1.1.7 (Alpha) variant, **d** the proportion of all performed PCR tests which turned out positive, **e** the overall level of non-pharmaceutical interventions as measured by the Oxford Coronavirus Government Response Tracker (OxCGRT) Stringency Index, and **f** the population shares having received at least one vaccination dose (dotted) and complete vaccination (solid). The dark gray area indicates the period addressed in the present manuscript, and the light gray area is the one from Bracher et al.[16].

To assess the full predictive distribution, we use the weighted interval score (WIS; ref. [22]). The WIS is a weighted average of interval scores at different nominal levels and the absolute error. For $N$ nested prediction intervals, it is defined as

$$\text{WIS}(F, y) = \frac{1}{2N+1} \times \left( |y - m| + \sum_{k=1}^{N} \left( \alpha_k \times \text{IS}_{\alpha_k}(F, y) \right) \right), \quad (2)$$

where $m$ is the predictive median and in our setting $N = 11$. The WIS is a well-known approximation of the continuous ranked probability score (CRPS; ref. [21]) and generalizes the absolute error to probabilistic forecasts. Its values can be interpreted on the natural scale of the data and measure how far the observed value $y$ is from the predictive distribution (lower values are thus better). For deterministic one-point forecasts, the WIS reduces to the absolute error. A useful property of the WIS is that it inherits the decomposition of the interval score into forecast spread, over-prediction, and underprediction, which makes average scores more interpretable. As secondary measures of forecast quality, we use the absolute error to assess the central tendency of forecasts

and interval coverage rates of 50 and 95% prediction intervals to assess calibration.

As specified in our study protocol, whenever forecasts from a model were missing for a given week, we imputed the score with the worst (largest) value achieved by any other model for the respective week and target. However, almost all teams provided complete sets of forecasts, and very few scores needed imputation.

**Submitted models and baselines.** During the evaluation period, forecasts from fifteen different models run by fourteen independent teams of researchers were collected. Thirteen of these were already available during the first part of our study, see Table 3 and Supplementary Note 3 of Bracher et al.[16] for detailed descriptions. Table 1 provides a slightly extended summary of model properties, including the two new models, itwm-dSEIR and Karlen-pypm; a more detailed description of the latter can be found in Supplement S1. All forecast data produced by teams was made available under open licenses. They do not contain any personal data, so no ethics approval was required for their use.

**Table 1 Forecast models contributed by independent external research teams.**

| Category | Model | NPI | Test | Variants | Age | DE | PL | Regional | Truth | Pr |
|---|---|---|---|---|---|---|---|---|---|---|
| Agent-based | ICM-agentModel[53] | ✓ | ✓ | ✓ | ✓ | | ✓ | | MZ | ✓ |
| | MOCOS-agent1[54] | ✓ | ✓ | ✓ | ✓ | | ✓ | | JHU | ✓ |
| Compartment | CovidAnalytics-DELPHI[55] | ✓ | | | | ✓ | ✓ | | JHU | ✓ |
| | FIAS_FZJ-Epi1Ger[56] | | | | | ✓ | | ✓ | RKI | ✓ |
| | itwm-dSEIR | | | | ✓ | ✓ | | | RKI | ✓ |
| | Karlen-pypm[57] | | | ✓ | | ✓ | | ✓ | RKI | ✓ |
| | LeipzigIMISE-SECIR[58] | ✓ | ✓ | ✓ | | ✓ | | (✓) | RKI | ✓ |
| | MIMUW-StochSEIR | | | | | | ✓ | | JHU | ✓ |
| | USC-SIkJalpha[59] | | | | | ✓ | ✓ | ✓ | RKI/MZ | ✓ |
| Growth rate/ renewal eq. | epiforecasts-EpiNow2[60] | | | | | ✓ | ✓ | ✓ | RKI/MZ | ✓ |
| | SDSC_ISG-TrendModel[61] | | | | | ✓ | ✓ | | JHU | |
| | ITWW-county_repro[62] | | | | ✓ | ✓ | ✓ | ✓ | RKI/MZ | ✓ |
| | LANL-GrowthRate[63] | | | | | ✓ | ✓ | | JHU | ✓ |
| Human judgment | epiforecasts-EpiExpert[64] | (✓) | (✓) | (✓) | (✓) | ✓ | ✓ | | RKI/MZ | ✓ |
| Forecast ensemble | Imperial-ensemble2[65] | | | | | ✓ | ✓ | | RKI | ✓ |

NPI: Does the forecast model explicitly account for non-pharmaceutical interventions? Test: Does the model account for changing testing strategies? Variants: Does the model accommodate multiple variants? Age: Is the model age-structured? DE, PL: Are forecasts issued for Germany and Poland, respectively? Regional: Were regional-level forecasts for at least one country submitted? Truth: Which truth data source does the model use? Pr: Are forecasts probabilistic (23 quantiles)? Detailed descriptions of the different models can be found in Bracher et al.[16], Supplementary Note 3, and in Supplementary Methods (Section 1) of this article.

During the evaluation period, only the ICM-agentModel explicitly accounted for vaccinations (given the low realized vaccination coverage by the end of the study period, this aspect likely had a limited impact). Only four models (ICM-agentModel, Karlen-pypm, LeipzigIMISE-SECIR, and MOCOS-agent1, all only for certain weeks) explicitly accounted for the presence of multiple variants. In contrast to other related projects[2], none of the models used mobility data or social media data.

To put the results achieved by the submitted models into perspective, the Forecast Hub team generated forecasts from three simple reference models (see also Bracher et al.[16], Supplementary Note 2). KIT-baseline is a simple last-observation-carried-forward model, i.e., it predicts the last observed value indefinitely into the future. Predictive quantiles are obtained by assuming a negative binomial observation model with a dispersion parameter estimated via maximum likelihood from five recent observations. KIT-extrapolation_baseline extrapolates exponential growth or decrease if the last three observations are monotonically increasing or decreasing, with a weekly growth rate equal to the one observed between the last and second last week; if the last three observations are not ordered, it predicts a plateau. Predictive quantiles are again obtained using a negative binomial observation model and five recent observations. KIT-time_series_baseline is an exponential smoothing time series model with multiplicative errors as used by ref. [23] to predict COVID-19 cases and deaths. It is implemented using the R package forecast, version 8.12[24].

As a further external comparison, we added publicly available death forecasts by the Institute for Health Metrics and Evaluation (IHME, University of Washington[25]; available under the CC BY-NC 4.0 license). Here, we always used the most recent prediction available on a given forecast date.

**Forecast ensembles**. The Forecast Hub team used the submitted forecasts to generate three different ensemble forecasts. In the KITCOVIDhub-median_ensemble, the $\alpha$-quantile of the ensemble forecast is obtained as the median of the $\alpha$-quantiles of the member forecasts. In the KITCOVIDhub-mean_ensemble, the mean instead of the median is applied for aggregation. In KIT-COVIDhub-inverse_wis_ensemble, a convex combination of the $\alpha$-quantiles of the member forecasts is used. The weights are chosen inversely proportional to the mean WIS value (see

equation (2)) obtained by the member models over the last six evaluated forecasts (last three 1-week-ahead, last two two-week-ahead, last three-week-ahead). This is done separately for each time series to be predicted. Missing scores are imputed by the worst score achieved by any model for the respective target, meaning that irregularly submitted models will be penalized and receive less weight.

In the study protocol, the median ensemble was defined as our primary ensemble approach[11] as it can be assumed to be more robust to occasionally misguided forecasts (e.g., due to technical errors). We therefore display this version in all figures and focus our discussion on it. Note that all forecast aggregations are performed directly at the level of quantiles rather than density functions as in other work[26]. This approach is referred to as Vincentization (in ref. [27], see e.g., ref. [28]). A broader discussion of Vincentization approaches and their application to epidemiological forecasts, including numerous other weighting schemes, can be found in recent works by Taylor and Taylor[29] and Ray et al.[30]. Notably, Taylor and Taylor[29] used a similar inverse score weighting approach and found it to perform well in a re-analysis of forecasts from the US COVID-19 Forecast Hub. In this context, we note that our inverse-WIS ensemble does not involve any estimation or optimization of weights, but simply uses the inverse of an average of past scores as heuristic weights. A more flexible approach with one tuning parameter estimated from the data has been used in Ray et al.[30].

There were no formal inclusion criteria other than the completeness of the submitted set of 23 quantiles. The Forecast Hub team did, however, occasionally exclude forecasts with a highly implausible central tendency or degree of dispersion manually. These exclusions have been documented in the Forecast Hub platform.

**Reporting summary**. Further information on research design is available in the Nature Research Reporting Summary linked to this article.

## Results

Figures 2 and 3 show the forecasts made by the median ensemble (KIT-median_ensemble; our prespecified main ensemble approach; see Materials and Methods); a naïve model always using the last observed value as the expectation for the following

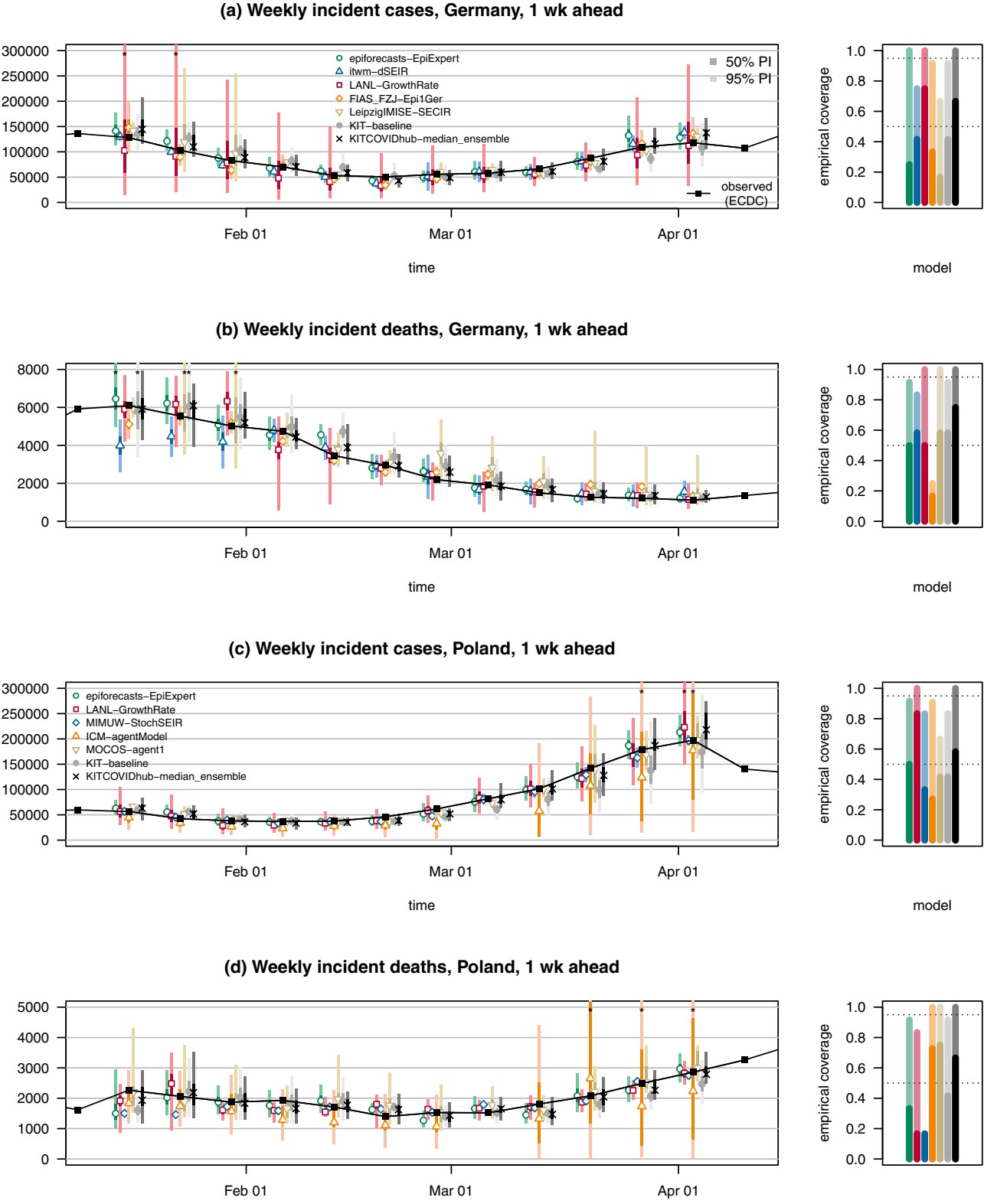

**Fig. 2 One-week-ahead forecasts of cases and deaths from COVID-19 in Germany and Poland.** One-week-ahead forecasts of confirmed cases and deaths from COVID-19 in Germany (**a**, **b**) and Poland (**c**, **d**). It shows forecasts from a baseline model, the median ensemble of all submissions, and a subset of submitted models with above-average performance. The black line shows observed data. Colored points represent predictive medians and dark and light bars show 50 and 95% prediction intervals, respectively. Asterisks mark intervals exceeding the upper plot limit. The remaining submitted models are displayed in Supplementary Fig. 1. The right column shows the empirical coverage rates of the different models. The dark and light bars represent the proportion of cases where the 50 and 95% prediction intervals, respectively, contained the observed values. The dotted lines show the desired nominal levels of 0.5 and 0.95.

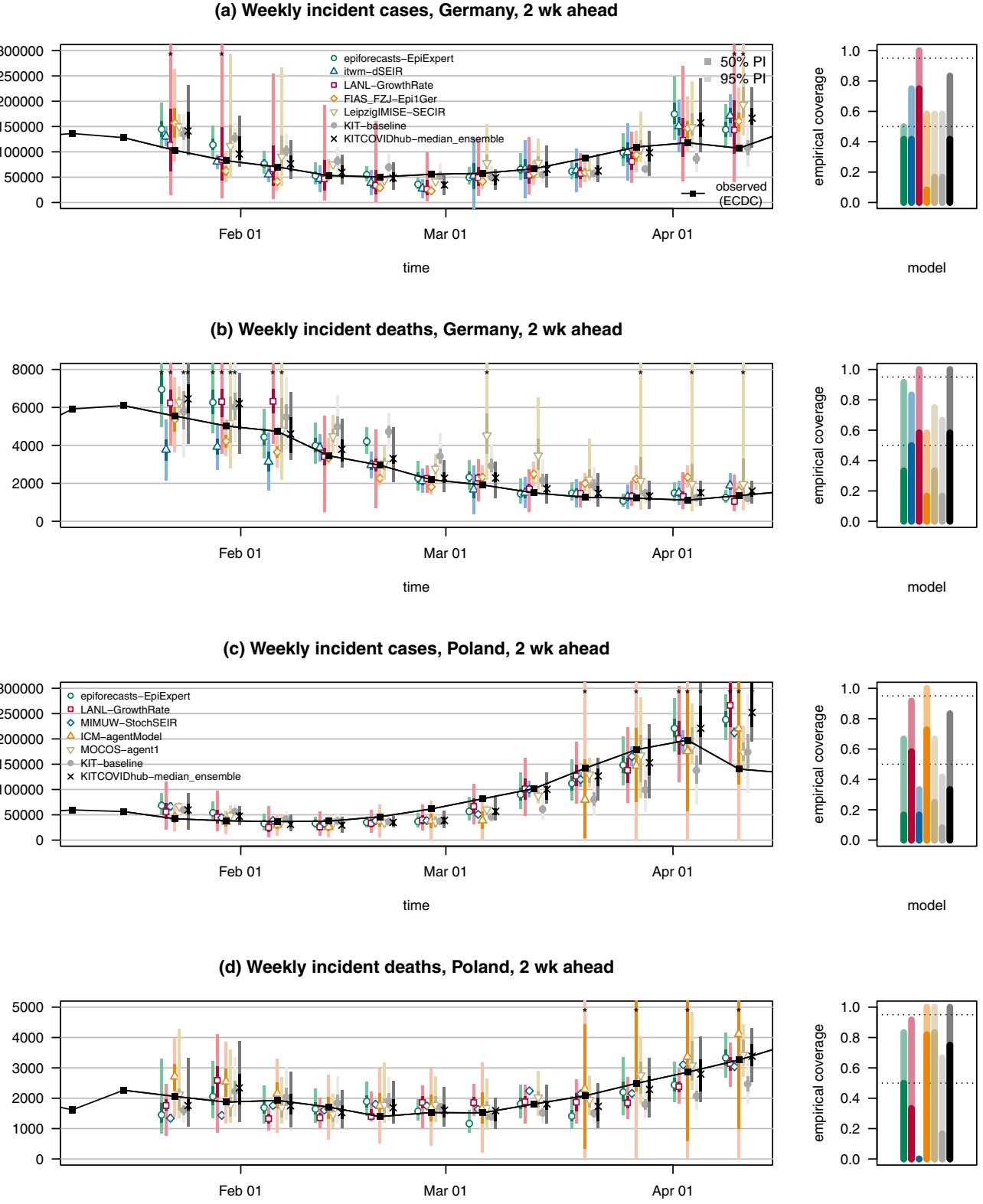

**Fig. 3 Two-week-ahead forecasts of cases and deaths from COVID-19 in Germany and Poland.** Two-week-ahead forecasts of confirmed cases and deaths from COVID-19 in Germany (**a**, **b**) and Poland (**c**, **d**). It shows forecasts from a baseline model, the median ensemble of all submissions, and a subset of submitted models. The remaining submitted models are displayed in Supplementary Fig. 2. The black line shows observed data. Colored points represent predictive medians, and dark and light bars show 50 and 95% prediction intervals, respectively. The right column shows the empirical coverage rates of the different models. See the caption of Fig. 2 for a detailed explanation of plot elements.

weeks (KIT-baseline); and five contributed models with above-average overall performance across locations and targets (i.e., quantities to be predicted). In each Figure, case and death forecasts for Germany are shown in panels (a) and (b), while the same for Poland is displayed in panels (c) and (d). The forecasts are probabilistic, and we display the 50% and 95% prediction intervals (PIs) along with the respective median. Forecasts by the remaining teams are illustrated in Supplementary Figs. 1, 2, and forecasts at horizons of three and 4 weeks are shown in Supplementary Figs. 3–6. In the following, we discuss the performance of these forecasts, starting with a formal statistical evaluation before directing attention to the behavior at inflection points.

**Formal evaluation, January–April 2021**. Table 2 and Fig. 4 (panels (a), (b) for Germany and (c), (d) for Poland) summarize the performance of the submitted baseline and ensemble models over the 12-week study period. Performance is measured via the average weighted interval score (WIS, see Methods section) and the mean absolute error of the predictive median. For both measures, lower values indicate better predictive performance. We here show the average scores on the absolute scale, where they can be interpreted as the average distance between the observed and predicted value (the WIS taking into account forecast uncertainty). A summary table of relative scores standardized by the performance of the naïve KIT-baseline model is available in Supplementary Table 1. The WIS can moreover be decomposed into components representing underprediction, forecast spread, and overprediction (see Methods), which we show in Supplementary Fig. 7. Detailed results in tabular form at horizons of three and 4 weeks ahead can be found in Supplementary Table 2. As specified in the study protocol, we also provide results for cumulative cases and deaths (Supplementary Tables 3, 4) and based on JHU rather than RKI/MZ data (Supplementary Tables 5, 6; evaluation against JHU data leads to slightly higher WIS and absolute errors, but quite similar relative performance of models). A graphical display of individual scores can be found in Supplementary Fig. 8.

Both for incident cases and deaths, a majority, but not all models outperformed the naïve baseline model KIT-baseline (a model outperforms the baseline for a given target whenever its bar in Fig. 4 does not reach into the gray area). As one would expect, the performance of all models considerably deteriorated for longer forecast horizons. The prespecified median ensemble was consistently among the best-performing methods, outperforming most individual forecasts for all targets. The KITCOVIDhub-inverse_wis_ensemble, which is an attempt to weigh member models based on recent performance, does not yield any clear benefits over the unweighted median and mean ensembles. As can be seen from Supplementary Figs. 9, 10, the weights fluctuate substantially, implying that the relative performance of different models may be too variable for performance-based weights to pay off. The KIT-extrapolation_baseline model shows quite reasonable relative performance for cases in both countries. Given the relatively long stretches of continued upward or downward trends in cases, this simple heuristic was not easy to beat and is rather close to the performance of the ensemble forecasts. For deaths, too, there are rather clear trends over the study period. Nonetheless, the different ensemble forecasts achieve substantial improvements over KIT-extrapolation_baseline, meaning that the deviations from the previous trends were predicted with some success.

The most striking cases of individual models outperforming the ensemble occurred for longer-range case forecasts in Poland. Here, the two microsimulation models, MOCOS-agent1 and ICM-agentModel performed considerably better. These two

models were arguably among the ones which were most meticulously tuned to the specific national context. It seems that this yielded benefits for longer horizons, while at shorter horizons, the ensemble and some considerably simpler models were at least on par (the best performance at the 1-week horizon being achieved by the compartmental model MIMUW-StochSEIR).

There were considerable differences in the forecast uncertainty of the different models. This can be seen from the quite variable forecast interval widths in Figs. 2, 3, and resulted in large differences in the empirical coverage rates of 50 and 95% prediction intervals (Table 2 and right column in the aforementioned figures). The ensemble methods performed quite favorably in terms of coverage, typically with slight undercoverage (i.e., prediction intervals cover the observations less frequently than intended) for cases and slight overcoverage (intervals cover more often than intended) for deaths. The differences in forecast dispersion are also reflected by the components of the weighted interval score shown in Supplementary Fig. 7 (see Materials and Methods for an explanation of the decomposition). Some models, most strikingly ITWW-county_repro, issued very sharp predictions, leading to very small dispersion components of the weighted interval score (the darkest block in the middle of the stacked bar). In turn, this model received rather large penalties for both over- and under-prediction. Other models, like LANL-GrowthRate, epiforecasts-EpiNow2, and ICM-agentModel issued comparatively wide forecasts, leading to WIS values with large dispersion components. While there is no clear rule on what the score decomposition of an ideal forecast should look like, comparisons of the components provide useful indications on how to improve a model (e.g., the ITWW-county_repro model might benefit from widening the uncertainty intervals).

A subset of models also provided forecasts at the subnational level (states in Germany, voivodeships in Poland). Table 3 provides a summary of the respective results at the 1 and 2-week horizons (results for three and four weeks can be found in Supplementary Table 7). Despite the rather low number of available models, the ensembles generally achieved improvements over the individual models and, with exceptions for case forecasts in Germany, clearly outperformed the baseline model KIT-baseline. The mean WIS values are lower for the regional forecasts than for the national-level forecasts in Table 2 primarily because the numbers to be predicted are lower at the regional level; the WIS—like the absolute error—scales with the order of magnitude of the predicted quantity and cannot be compared directly across different forecasting tasks. Coverage of the ensemble forecasts was close to the nominal level for deaths and somewhat lower for cases. Note that in this comparison, part of the forecasts from the FIAS_FZJ-epi1Ger model were created retrospectively (using only the data available up to the forecast date) as the team only started issuing forecasts for all German federal states on 22 February 2021.

As specified in the study protocol[11], we also report evaluation results at the national level pooled across the two study periods for those models which covered both. These are summarized in Supplementary Tables 8, 9. For deaths, ensemble forecasts clearly outperformed individual models, the four-week-ahead horizon in Poland being the only one at which an individual model (epiforecasts-EpiExpert) meaningfully outperformed the prespecified median ensemble. While most contributed and baseline models were somewhat overconfident, the ensemble showed close to nominal coverage even at the 4-week-ahead horizon. For cases, the median ensemble achieved good relative performance (comparable to the best individual models) one and 2 weeks ahead, but was outperformed by a number of other models at 3 and 4 weeks. Notably, it failed to beat the naïve last-observation-

**Table 2 Forecast evaluation for Germany and Poland (incidence scale, based on RKI/MZ data).**

### Germany

| Model | 1 wk ahead cases | | | | 2 wk ahead cases | | | | 1 wk ahead deaths | | | | 2 wk ahead deaths | | | |
|---|---|---|---|---|---|---|---|---|---|---|---|---|---|---|---|---|
| | AE | WIS | $C_{0.5}$ | $C_{0.95}$ | AE | WIS | $C_{0.5}$ | $C_{0.95}$ | AE | WIS | $C_{0.5}$ | $C_{0.95}$ | AE | WIS | $C_{0.5}$ | $C_{0.95}$ |
| epiforecasts-EpiExpert | 9252 | 5415 | 0.25 | 1.00 | 20,233 | 13,607 | 0.42 | 0.50 | 300 | 204 | 0.50 | 0.92 | 509 | 323 | 0.33 | 0.92 |
| epiforecasts-EpiNow2 | 9676 | 6644 | 0.67 | 0.83 | 29,348 | 21,478 | 0.58 | 0.67 | 300 | 188 | 0.75 | 0.92 | 581 | 417 | 0.67 | 0.75 |
| FIAS_FZJ-Epi1Ger | 10,218 | 6294 | 0.33 | 0.92 | 25,662 | 16,621 | 0.08 | 0.58 | 436 | 336 | 0.17 | 0.25 | 655 | 475 | 0.17 | 0.58 |
| IHME-CurveFit | | | | | | | | | 516 | | | | 656 | | | |
| Imperial-ensemble2 | *23,011 | *15,824 | 0.27 | 0.91 | | | | | *193 | *136 | 0.80 | 0.90 | | | | |
| itwm-dSEIR | 6905 | 4644 | 0.42 | 0.75 | 18,935 | 13,626 | 0.42 | 0.75 | 483 | 326 | 0.58 | 0.83 | 534 | 354 | 0.50 | 0.83 |
| ITWW-county_repro | 15,223 | 12,418 | 0.08 | 0.25 | 31,836 | 25,851 | 0.00 | 0.17 | 564 | 527 | 0.00 | 0.00 | 286 | 236 | 0.08 | 0.25 |
| Karlen-pypm | 18,532 | 13,629 | 0.50 | 0.92 | 35,010 | 25,385 | 0.25 | 0.83 | 380 | 232 | 0.42 | 0.92 | 628 | 394 | 0.08 | 0.83 |
| LANL-GrowthRate | 12,623 | 10,542 | 0.75 | 1.00 | 15,797 | 13,945 | 0.75 | 1.00 | 338 | 222 | 0.50 | 1.00 | 425 | 265 | 0.58 | 1.00 |
| LeipzigIMISE-SECIR | 9161 | 6376 | 0.17 | 0.67 | 26,650 | 19,185 | 0.17 | 0.58 | 370 | 281 | 0.58 | 1.00 | 874 | 636 | 0.33 | 0.75 |
| MIT_CovidAnalytics-DELPHI | *11,910 | *8277 | 0.55 | 0.91 | *22,734 | *16,006 | 0.36 | 0.73 | 803 | 490 | 0.42 | 0.92 | 773 | 451 | 0.58 | 1.00 |
| SDSC_ISG-TrendModel | 7861 | | | | | | | | 436 | | | | | | | |
| USC-SIkJalpha | 13,766 | 9001 | 0.25 | 0.83 | 25,730 | 17,681 | 0.17 | 0.58 | 381 | 255 | 0.50 | 0.83 | 568 | 348 | 0.25 | 0.83 |
| KIT-baseline | 12,756 | 7953 | 0.42 | 0.92 | 23,785 | 17,330 | 0.17 | 0.58 | 411 | 277 | 0.58 | 0.92 | 780 | 525 | 0.17 | 0.67 |
| KIT-extrapolation_baseline | 8823 | 5715 | 0.50 | 1.00 | 22,858 | 14,679 | 0.33 | 0.75 | 456 | 269 | 0.33 | 1.00 | 806 | 490 | 0.33 | 0.83 |
| KIT-time_series_baseline | 15,583 | 10,281 | 0.25 | 0.75 | 32,306 | 22,026 | 0.25 | 0.67 | 406 | 263 | 0.67 | 1.00 | 851 | 601 | 0.50 | 0.92 |
| KITCOVIDhub-inverse_wis_ensemble | 8586 | 5294 | 0.58 | 1.00 | 22,000 | 13,824 | 0.50 | 0.83 | 216 | 149 | 0.75 | 1.00 | 307 | 207 | 0.75 | 1.00 |
| KITCOVIDhub-mean_ensemble | 8377 | 5277 | 0.75 | 1.00 | 21,825 | 13,662 | 0.50 | 0.92 | 220 | 152 | 0.58 | 1.00 | 346 | 219 | 0.75 | 1.00 |
| KITCOVIDhub-median_ensemble | 7344 | 4660 | 0.67 | 1.00 | 19,296 | 12,734 | 0.42 | 0.83 | 232 | 150 | 0.75 | 1.00 | 376 | 225 | 0.58 | 1.00 |

### Poland

| Model | 1 wk ahead cases | | | | 2 wk ahead cases | | | | 1 wk ahead deaths | | | | 2 wk ahead deaths | | | |
|---|---|---|---|---|---|---|---|---|---|---|---|---|---|---|---|---|
| | AE | WIS | $C_{0.5}$ | $C_{0.95}$ | AE | WIS | $C_{0.5}$ | $C_{0.95}$ | AE | WIS | $C_{0.5}$ | $C_{0.95}$ | AE | WIS | $C_{0.5}$ | $C_{0.95}$ |
| epiforecasts-EpiExpert | 7500 | 4553 | 0.50 | 0.92 | 25,316 | 17,408 | 0.17 | 0.67 | 208 | 137 | 0.33 | 0.92 | 287 | 181 | 0.50 | 0.83 |
| epiforecasts-EpiNow2 | 7928 | 5906 | 0.58 | 0.92 | 29,762 | 22,098 | 0.42 | 0.83 | 184 | 119 | 0.58 | 1.00 | 340 | 228 | 0.58 | 1.00 |
| ICM-agentModel | *23,011 | *15,824 | 0.27 | 0.91 | *26,694 | *18,098 | 0.73 | 1.00 | *488 | *294 | 0.73 | 1.00 | *605 | *507 | 0.82 | 1.00 |
| IHME-CurveFit | | | | | | | | | 374 | | | | 520 | | | |
| Imperial-ensemble2 | | | | | | | | | *188 | *138 | 0.30 | 0.70 | | | | |
| ITWW-county_repro | 20,054 | 17,364 | 0.17 | 0.25 | 36,651 | 31,445 | 0.17 | 0.50 | 589 | 551 | 0.00 | 0.00 | 784 | 711 | 0.00 | 0.00 |
| LANL-GrowthRate | 8129 | 5787 | 0.83 | 1.00 | 23,269 | 15,240 | 0.58 | 0.92 | 229 | 137 | 0.17 | 0.83 | 347 | 216 | 0.33 | 0.92 |
| MIMUW-StochSEIR | 5705 | 4028 | 0.33 | 0.83 | 17,642 | 15,347 | 0.17 | 0.33 | 237 | 224 | 0.17 | 0.17 | 288 | 267 | 0.00 | 0.00 |
| MOCOS-agent1 | *22,344 | *12,912 | 0.20 | 0.90 | *49,687 | *33,033 | 0.10 | 0.70 | *393 | *244 | 0.55 | 1.00 | *520 | *296 | 0.36 | 1.00 |
| SDSC_ISG-TrendModel | 5173 | 4978 | 0.42 | 0.67 | 15,022 | 11,380 | 0.25 | 0.67 | 158 | 132 | 0.75 | 1.00 | 203 | 149 | 0.83 | 1.00 |
| USC-SIkJalpha | 6323 | 6919 | 0.33 | 0.83 | 32,822 | 24,436 | 0.17 | 0.50 | 265 | 133 | 0.33 | 0.92 | 266 | 168 | 0.42 | 0.92 |
| KIT-baseline | 10,404 | 9736 | 0.42 | 0.83 | 32,182 | 22,709 | 0.08 | 0.42 | 206 | 167 | 0.42 | 0.92 | 416 | 275 | 0.17 | 0.67 |
| KIT-extrapolation_baseline | 16,407 | 5992 | 0.50 | 0.92 | 29,638 | 22,165 | 0.25 | 0.58 | 258 | 190 | 0.58 | 0.83 | 404 | 284 | 0.42 | 0.83 |
| KIT-time_series_baseline | 9448 | 7787 | 0.75 | 0.83 | 30,359 | 21,510 | 0.50 | 0.75 | 269 | 232 | 0.67 | 0.67 | 467 | 362 | 0.50 | 0.58 |
| KITCOVIDhub-inverse_wis_ensemble | 10,784 | 4689 | 0.50 | 1.00 | 23,418 | 15,580 | 0.42 | 0.83 | 150 | 111 | 0.75 | 1.00 | 197 | 144 | 0.75 | 1.00 |
| KITCOVIDhub-mean_ensemble | 6866 | 4784 | 0.75 | 1.00 | 23,673 | 15,573 | 0.33 | 0.83 | 141 | 114 | 0.75 | 1.00 | 173 | 152 | 0.92 | 1.00 |
| KITCOVIDhub-median_ensemble | 7130 | 4403 | 0.58 | 1.00 | 23,027 | 16,241 | 0.33 | 0.83 | 162 | 103 | 0.67 | 1.00 | 193 | 137 | 0.75 | 1.00 |

Summaries are based on 12 weekly forecasts per target.
*Abbreviations:* $C_{0.5}$, $C_{0.95}$: coverage rates of the 50 and 95% prediction intervals, AE: mean absolute error, WIS: mean weighted interval score.
*Asterisks mark entries where scores were imputed for at least one week. Weighted interval scores and absolute errors were imputed with the worst (largest) score achieved by any other forecast for the respective target and week. Models marked thus received a pessimistic assessment of their performance. If a model covered less than two-thirds of the evaluation period, results are omitted.

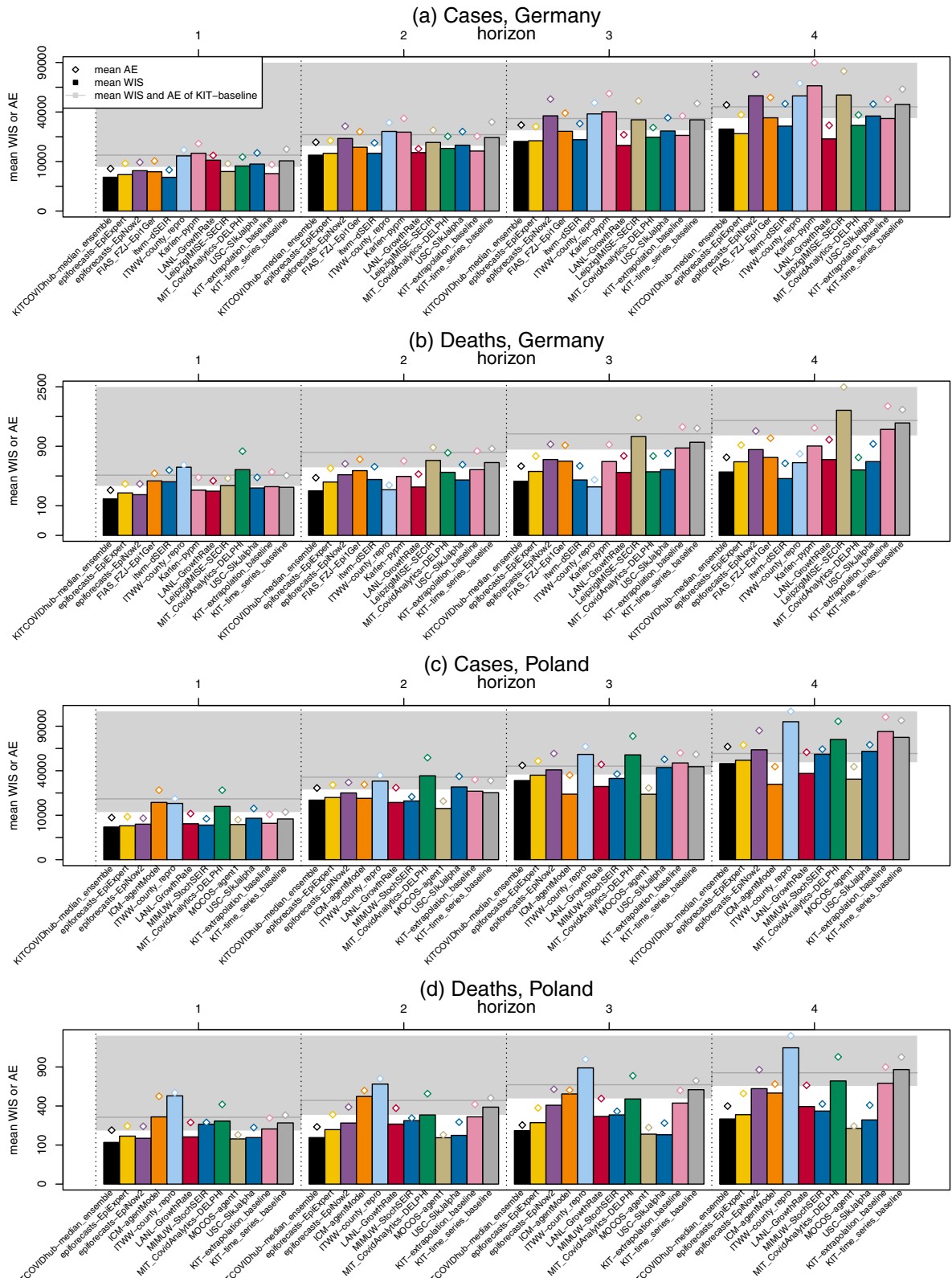

**Fig. 4 Formal evaluation results in terms of mean weighted interval scores.** Average weighted interval scores (bars) and absolute errors (diamonds) achieved by models in Germany (**a**, **b**) and Poland (**c**, **d**) per target and forecast horizon (12 weekly forecasts). The bottom end of the gray area represents the mean WIS of the baseline model KIT-baseline, and the gray horizontal line is its mean absolute error. Values are shown on a square-root scale to enhance readability. Only models covering all four horizons are shown.

**Table 3 Forecast evaluation at the regional level, Germany and Poland (incidence scale, RKI/MZ data).**

**Germany**

| Model | 1 wk ahead cases | | | | 2 wk ahead cases | | | | 1 wk ahead deaths | | | | 2 wk ahead deaths | | | |
|---|---|---|---|---|---|---|---|---|---|---|---|---|---|---|---|---|
| | AE | WIS | $C_{0.5}$ | $C_{0.95}$ | AE | WIS | $C_{0.5}$ | $C_{0.95}$ | AE | WIS | $C_{0.5}$ | $C_{0.95}$ | AE | WIS | $C_{0.5}$ | $C_{0.95}$ |
| epiforecasts-EpiNow2 | 694 | 448 | 0.52 | 0.90 | 1663 | 1120 | 0.39 | 0.77 | 36 | 24 | 0.54 | 0.92 | 65 | 43 | 0.39 | 0.84 |
| FIAS_FZJ-Epi1Ger | 801 | 518 | 0.30 | 0.76 | 1732 | 1173 | 0.25 | 0.66 | 41 | 34 | 0.11 | 0.30 | 54 | 41 | 0.12 | 0.45 |
| IHME-CurveFit | | | | | | | | | 47 | 34 | 0.15 | 0.46 | 53 | 20 | 0.26 | 0.69 |
| ITWW-county_repro | 996 | 671 | 0.36 | 0.73 | 2053 | 1392 | 0.40 | 0.78 | 41 | 22 | 0.62 | 0.94 | 28 | 28 | 0.47 | 0.95 |
| Karlen-pypm | 1251 | 804 | 0.65 | 0.95 | 2300 | 1432 | 0.48 | 0.92 | 35 | 21 | 0.08 | 0.42 | 48 | 28 | 0.08 | 0.42 |
| LeipzigIMISE-SECIR | 941 | 631 | 0.08 | 0.58 | 1614 | 1094 | 0.17 | 0.33 | | | | | | | | |
| USC-SIkJalpha | 859 | 544 | 0.45 | 0.86 | 1536 | 1085 | 0.32 | 0.74 | 32 | 25 | 0.46 | 0.91 | 43 | 38 | 0.33 | 0.83 |
| KIT-baseline | 785 | 504 | 0.42 | 0.89 | 1616 | 1073 | 0.28 | 0.65 | 38 | 31 | 0.51 | 0.92 | 57 | 50 | 0.35 | 0.78 |
| KIT-extrapolation_baseline | 1010 | 665 | 0.47 | 0.89 | 1954 | 1320 | 0.34 | 0.76 | 49 | 36 | 0.46 | 0.88 | 76 | 59 | 0.43 | 0.85 |
| KIT-time_series_baseline | 868 | 523 | 0.40 | 0.80 | 1744 | 1098 | 0.28 | 0.68 | 59 | | 0.36 | 0.92 | 93 | | 0.33 | 0.84 |
| KITCOVIDhub-inverse_wis_ensemble | 848 | 515 | 0.49 | 0.97 | 1693 | 1044 | 0.43 | 0.89 | 28 | 18 | 0.56 | 0.97 | 39 | 25 | 0.50 | 0.94 |
| KITCOVIDhub-mean_ensemble | | | 0.50 | 0.97 | | | | 0.91 | 28 | 18 | 0.56 | 0.97 | 41 | 26 | 0.51 | 0.96 |
| KITCOVIDhub-median_ensemble | 769 | 485 | 0.57 | 0.95 | 1661 | 1028 | 0.42 | 0.90 | 29 | 19 | 0.57 | 0.94 | 36 | 23 | 0.49 | 0.95 |

**Poland**

| Model | 1 wk ahead cases | | | | 2 wk ahead cases | | | | 1 wk ahead deaths | | | | 2 wk ahead deaths | | | |
|---|---|---|---|---|---|---|---|---|---|---|---|---|---|---|---|---|
| | AE | WIS | $C_{0.5}$ | $C_{0.95}$ | AE | WIS | $C_{0.5}$ | $C_{0.95}$ | AE | WIS | $C_{0.5}$ | $C_{0.95}$ | AE | WIS | $C_{0.5}$ | $C_{0.95}$ |
| epiforecasts-EpiNow2 | 632 | 426 | 0.58 | 0.89 | 2081 | 1500 | 0.41 | 0.78 | 27 | 18 | 0.49 | 0.86 | 47 | 33 | 0.44 | 0.82 |
| ITWW-county_repro | 1351 | 956 | 0.36 | 0.70 | 2532 | 1867 | 0.40 | 0.74 | 45 | 38 | 0.12 | 0.31 | 56 | 44 | 0.19 | 0.42 |
| USC-SIkJalpha | 853 | 622 | 0.40 | 0.90 | 1841 | 1305 | 0.25 | 0.64 | 20 | 13 | 0.51 | 0.95 | 27 | 18 | 0.45 | 0.90 |
| KIT-baseline | 1085 | 647 | 0.34 | 0.84 | 2028 | 1451 | 0.15 | 0.52 | 26 | 17 | 0.51 | 0.91 | 34 | 23 | 0.36 | 0.79 |
| KIT-extrapolation_baseline | 702 | 457 | 0.47 | 0.89 | 2069 | 1485 | 0.34 | 0.70 | 29 | 19 | 0.46 | 0.89 | 39 | 25 | 0.43 | 0.89 |
| KIT-time_series_baseline | 804 | 562 | 0.53 | 0.83 | 2123 | 1459 | 0.42 | 0.73 | 32 | 21 | 0.41 | 0.80 | 44 | 31 | 0.32 | 0.70 |
| KITCOVIDhub-inverse_wis_ensemble | 658 | 399 | 0.60 | 0.98 | 1786 | 1155 | 0.43 | 0.82 | 19 | 12 | 0.53 | 0.93 | 25 | 16 | 0.51 | 0.93 |
| KITCOVIDhub-mean_ensemble | 604 | 380 | 0.61 | 0.99 | 1663 | 1061 | 0.43 | 0.88 | 19 | 13 | 0.53 | 0.93 | 26 | 17 | 0.52 | 0.94 |
| KITCOVIDhub-median_ensemble | 615 | 385 | 0.64 | 0.97 | 1801 | 1125 | 0.42 | 0.83 | 20 | 13 | 0.56 | 0.93 | 26 | 17 | 0.50 | 0.94 |

Results are averaged over the different regions (16 states in Germany, 16 voivodeships in Poland).
*Abbreviations:* $C_{0.5}$, $C_{0.95}$: coverage rates of the 50 and 95% prediction intervals, AE: mean absolute error, WIS: mean weighted interval score.
*Asterisks mark entries where scores were imputed for at least 1 week. Weighted interval scores and absolute errors were imputed with the worst (largest) score achieved by any other forecast for the respective target and week. Models marked thus received a pessimistic assessment of their performance. If a model covered less than two-thirds of the evaluation period, results are omitted.

carried-forward model KIT-baseline. Its coverage of prediction intervals was acceptable 1 week ahead, but substantially below nominal at higher horizons (e.g., 13/19 and 10/19 four weeks ahead in Germany and Poland, respectively, at the 0.95 level), which reflects the severe difficulties in predicting cases in Fall 2020 as discussed in ref. [16].

**Behavior at inflection points**. From a public health perspective, there is often a specific interest in how well models anticipate major inflection points (changes in trend). We therefore discuss these instances separately. However, we note that, as will be detailed in the discussion, post-hoc conditioning of evaluation results on the occurrence of unusual events comes with important conceptual challenges.

The renewed increase in cases in both Germany and Poland (third wave) in late February 2021 was due to the shift from the wild-type variant of the virus to the B.1.1.7 (or Alpha) variant, see Fig. 1c for estimated shares of the new variant over time. Given earlier observations about the spread of the B.1.1.7 variant in the UK[31] and Denmark, there was public discussion about the likelihood of a re-surgence, but there was considerable uncertainty about the timing and strength (see e.g., a German newspaper article[32] from early February 2021). This was largely due to the limited availability of representative sequencing data. In certain regions of Germany, specifically the city of Cologne[33] and the state of Baden-Württemberg[34], large-scale sequencing

had been adopted by late January, but results were considered difficult to extrapolate to the whole of Germany. An updated RKI report[35] on virus variants from 10 February 2020 described a "continuous increase in the share of the VOC B.1.1.7", but cautioned that the data were "subject to biases, e.g., with respect to the selection of samples to sequence" (our translation).

Given the limited available data, and the fact that many approaches had not been designed to accommodate multiple variants, only two of the teams submitting forecasts for Germany opted to account for this aspect (a question which was repeatedly discussed during coordination calls). These exceptions were the Karlen-pypm and LeipzigIMISE-SECIR models, which starting from 1 March 2021, explicitly accounted for the presence of two variants. As a result, most models did not anticipate the change in trend well and only reacted implicitly once the change became apparent in the data on 27 February 2021. Figure 5 shows the case forecasts of all submitted models and the median ensemble from (a) 15 February, (b) 22 February, and (c) 1 March 2021. In panel (d), we moreover show the two short time series of shares of the B.1.1.7 variant available from Robert Koch Institute at the respective prediction time points.

The ITWW-county_repro model was the only one to anticipate a change in trend on 15 February (though slower than the observed one) and adapted quickly to the upward trend in the following week. This model extrapolates recently observed growth or decline at the county level and aggregates these fine-grained

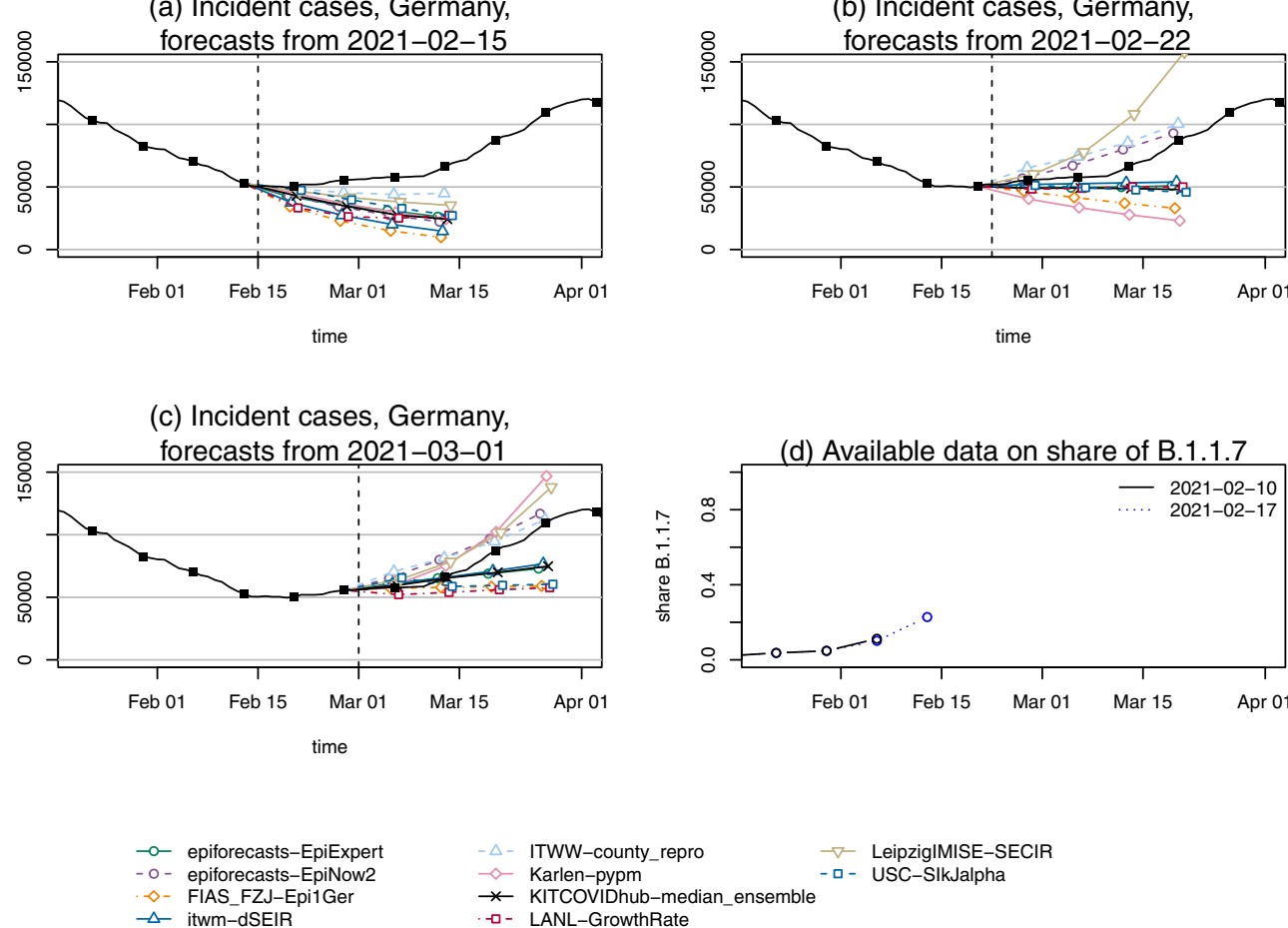

**Fig. 5 Case forecasts in Germany preceding the upward trend change in March 2022.** Point forecasts of cases in Germany, as issued on **a** 15 February, **b** 22 February, and **c** 1 March 2021. These dates, shown as vertical dashed lines, mark the start of a renewed increase in overall case counts due to the new variant of concern B.1.1.7. **d** Data by RKI on the share of the B.1.1.7 variant as available on the different forecast dates (the next data release by RKI occurred on 3 March). The models Karlen-pypm and LeipzigIMISE-SECIR accounted for the presence of multiple variants from 1 March onwards.

forecasts to the state or national level. Therefore, it may have been able to catch a signal of renewed growth, as a handful of German states had already experienced a slight increase in cases in the previous week (e.g., Thuringia and Saxony-Anhalt, see panel (b) of Supplementary Fig. 11). However, as illustrated in panel (a) of the same Figure, the ITWW model had also predicted turning points earlier during the same phase of decline in cases, and might generally have a tendency to produce such patterns. Another noteworthy observation in this context is the change in the predictions of the Karlen-pypm model. After the extension of the model to account for the B.1.1.7 variant on 1 March, its forecasts changed from the most optimistic to the most pessimistic among all included models (panels b and c of Fig. 5). The other model including variant data, LeipzigIMISE-SECIR, likewise was among the first to adopt an upward trend.

In Poland, the availability of sequencing data was very limited during our study period; the GISAID database[19] only contained 2271 sequenced samples for Poland by 29 March 2021[18]. Nonetheless, the ICM-agentModel and MOCOS-agent1 models explicitly took the presence of a new variant into account to the degree possible. Again, the ITWW-county_repro model was the first to predict a change in overall trends (in this case, without having predicted turning points already in the preceding weeks; see Supplementary Fig. 1).

In Poland, the third wave reached its peak in the week ending on 3 April 2021. Despite the fact that it coincided with the Easter

weekend and thus somewhat unclear data quality, this turn-around was predicted quite well by two Poland-based teams, MOCOS-agent1 and ICM-agentModel. Figure 6 shows forecasts made on (a) 22 March, (b) 29 March, and (c) 5 April. It can be seen that the trajectory predicted by the two mentioned models differed substantially from those of most others, including the ensemble, which predicted a sustained increase. This successful prediction of the turning point was in large part responsible for the good relative performance of MOCOS-agent1 and ICM-agentModel at longer horizons (Table 2). In retrospective discussions, the respective teams noted that the tightening of non-pharmaceutical interventions (NPIs) on 27 March (which they had anticipated) in combination with possible seasonal effects had led them to expect a downward turn.

For Germany, the peak of the third wave occurred only after the end of our prespecified study period, but we note that numerous models showed strong overshoot as they expected the upward trend to continue. The exact mechanisms underlying the turnaround remain poorly understood. A new set of restrictions referred to as the Bundesnotbremse in German (federal emergency break) was introduced too late to explain the change on its own.

In Germany, the study period coincided almost perfectly with a prolonged period of decline in deaths. In Fig. 7, panels (a) and (b) show the behavior of the median ensemble at the beginning and end of this phase. The ensemble had already anticipated a

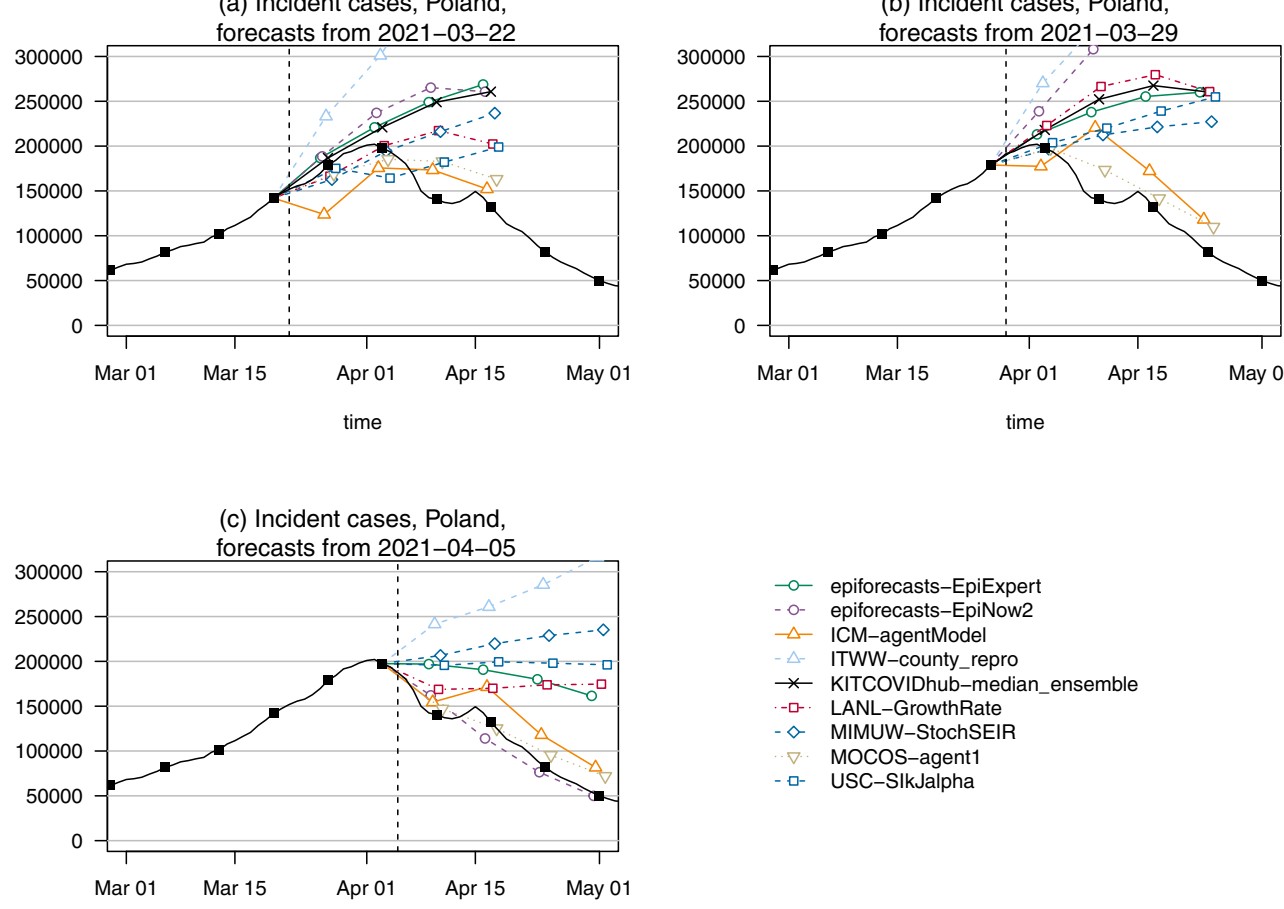

**Fig. 6 Case forecasts in Poland surrounding the peak in April 2022.** Point forecasts of cases in Poland from **a** 22 March, **b** 29 March, and **c** 5 April 2021, surrounding the peak week. In each panel, the date at which forecasts were created is marked by a dashed vertical line. The models ICM-agentModel and MOCOC-agent1 anticipated the trend change correctly, while the remaining models show more or less pronounced overshoot.

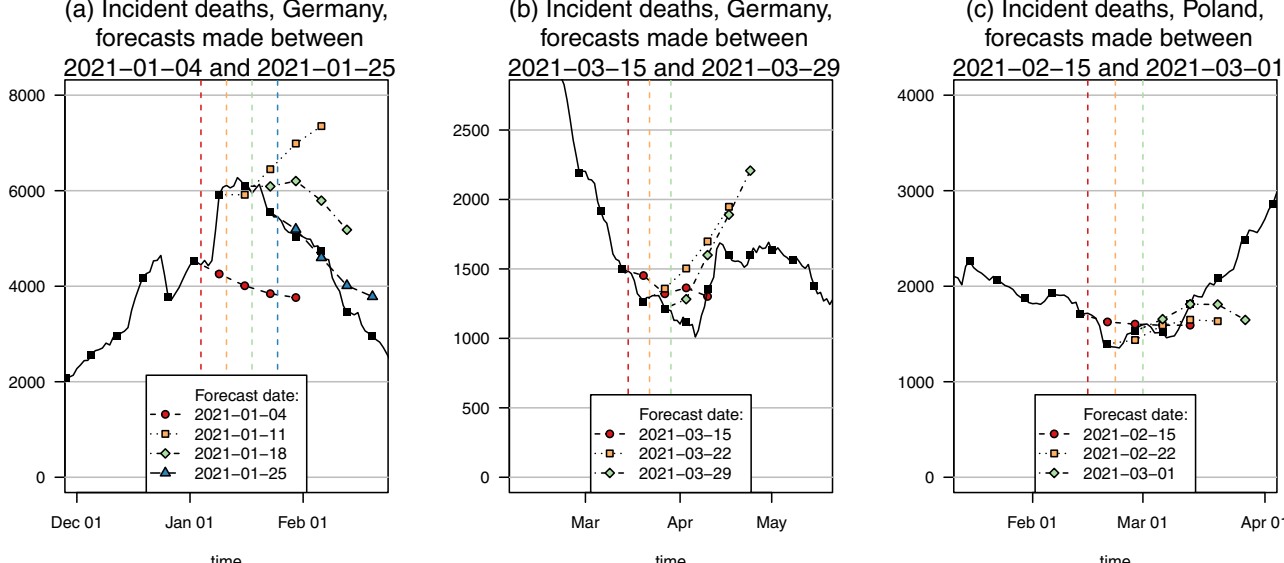

**Fig. 7 Death forecasts preceding trend changes.** Point forecasts of the median ensemble during changing trends in deaths. **a** Downward turn in Germany, January 2021. **b** Upward turn in Germany, March 2021. **c** Upward turn in Poland, February/March 2021. Different colors and point/line shapes represent forecasts made at distinct time points (marked by dashed vertical lines).

downward turn on 4 January, two weeks before it actually occurred. Following the unexpectedly strong increase in the following week, it went to extending the upward tendency, before switching back to predicting a turnaround. It seems likely that the irregular pattern in late December and early January is partly due to holiday effects in reporting, and forecast models may have been disturbed by this aspect.

At the end of the downward trend in late March, the ensemble again anticipated the turnaround to arrive earlier than it did, and predicted a more prolonged rise than was observed. Nonetheless, in both cases, the ensemble, to some degree, anticipated qualitative change, and the observed trajectories were well inside the respective 95% prediction intervals (with the exception of the forecast from 4 January; however, this forecast had prospectively been excluded from the analysis as we anticipated reporting irregularities).

In Poland, deaths started to increase in early March after a prolonged period of decay. As can be seen in panel (c) of Fig. 7, the median ensemble had anticipated this change (22 February 2021), but in terms of its point forecast, did not initially expect a prolonged upward trend as later observed. Nonetheless, the observed trajectory was contained in the relatively wide 95% prediction intervals (Figs. 2, 3).

## Discussion

We presented results from the second and final part of a pre-registered forecast evaluation study conducted in Germany and Poland (January–April 2021). During the period covered in this paper, ensemble approaches yielded very good performance relative to contributed individual models and baseline models. The majority of contributed models was able to outperform a simple last-observation-carried-forward model for most targets and forecast horizons up to four weeks.

The results in this manuscript differ in important aspects from those for our first evaluation period (October–December 2020), when most models struggled to meaningfully outperform the KIT-baseline model for cases. Fall 2020 was characterized by rapidly changing non-pharmaceutical intervention measures, making it hard for models to anticipate the case trajectory. Pooled across both study periods, we found ensemble forecasts of deaths

to yield satisfactory reliability and clear improvements over baseline models. For cases, however, coverage was clearly below nominal from the two-week horizon onward, and in terms of mean weighted interval scores, the ensemble failed to outperform the KIT-baseline model three and four weeks ahead. This strengthens our previous conclusion[16] that meaningful case forecasts are only feasible at very short horizons. It also agrees with recent results from the US COVID-19 Forecast Hub[36], which led the organizers to temporarily suspend ensemble case forecasts beyond the 1-week horizon.

The differences between our two study periods illustrate that performance relative to simple baseline models is strongly dependent on how good a fit these are for a given period. Cases in Germany plateaued during November and early December 2020, making the last-observation-carried-forward strategy of KIT-baseline difficult to beat. The second evaluation period was characterized by longer stretches of continued upward or downward trends, making it much easier to beat that baseline. In this situation, however, many models did not achieve strong improvements over the extrapolation approach KIT-extrapolation_baseline. Ideally, one would wish complex forecast models to outperform each of these different baseline models. However, there are many ways of specifying a simple baseline[37], and post-hoc at least one of them will likely be in acceptable agreement with the observed trajectory. While the choice of the most meaningful reference remains subject to debate, we believe that the use of a small set of prespecified baselines as in the present study is a reasonable approach.

An observation made for both the first and the second part of our study is that predicting changing trends in cases is very challenging; turnarounds in death counts are less difficult to anticipate. This finding is shared by works on real-time forecasts of COVID-19 in the UK[38] and the US[39]. To interpret these insights, we note that, in principle, there are two ways of forecasting epidemiological time series. The first approach is to apply a mechanistic model to project future spread based on recent trends and other relevant factors like NPIs, population behavior, spread of different variants, or vaccination. Models can then predict trend changes based on classical epidemiological mechanisms (depletion of susceptibles) or observed/anticipated changes in surrounding factors, which depending on the model,

may be treated as exogenous or endogenous. The second approach is to establish a statistical relationship (often with a mechanistic motivation) to a leading indicator, i.e., a data stream which is informative on the trajectory of the quantity of interest, but available earlier. Changes in the trend of the leading indicator can then help anticipate future turning points in the time series of interest.

Death forecasts belong to the second category, with cases and hospitalizations serving as leading indicators. This prediction task has been addressed with considerable success. Case forecasts, on the other hand, typically are based on the first approach, which largely reduces trend extrapolation, unless models are carefully tuned to changing NPIs (see Table 1). Theoretical arguments on the limited predictability of turning points in such curves have been brought forward[40,41], and empirical work including ours confirms that this is a very difficult task. While the success of the two microsimulation models MOCOS-agent1 and ICM-agentModel in anticipating the downward turn in cases in Poland remains a rather rare exception, it shows that careful mechanistic modeling combined with in-depth knowledge of national specificities has the potential to anticipate the impact of changing NPIs. As both groups heavily drew from experience from past NPIs in Poland, there is hope that predictions of the effects of NPIs will further improve as experience accumulates. An alternative strategy to improve case forecasts would be to identify appropriate leading indicators. These could, for instance, be trajectories in other countries[42] or additional data streams e.g., mobility, insurance claims, or web searches. However, the benefits of such data for short-term forecasting thus far have been found to be modest[43]. Changes in dominant variants may make changes in overall trends predictable as they arise from the superposition of adverse but stable trends for the different variants. The availability of sequencing data has improved considerably since our study period, and we consider the extension of models to accommodate multiple strains a key step towards improved prediction of trend changes. Other relevant aspects include seasonal effects, which during our study period remained poorly understood due to limited historical data, and population immunity. As more data on seroprevalence become available, predictability of saturation effects may increase, though this will likely be complicated by the further evolution of the pathogen.

Another difficulty of case forecasts is incomplete case ascertainment, which must be assumed to vary over time[9,44]. As a consequence, data can be difficult to compare across different phases of the pandemic, and modelers often choose to only use a recent subset of the available data to calibrate their models. While data on testing volumes and test positivity rates are available, estimation of the reporting fractions and anticipation of its future development is challenging. Even if models correctly reflect underlying epidemic dynamics, this may thus not translate to accurate forecasts of the observed number of confirmed cases. This is a limitation of the considered forecasts and their evaluation, which inherit the difficulties of the underlying truth data sources. Nonetheless, we argue that a distinguishing feature of forecasts is that they refer to observable quantities, and forecasters should take into account all relevant aspects of the system producing them. Indeed, many of the considered models (e.g., MOCOS-agent1 and FIAS_FZJ-Epi1Ger) attempt to reconstruct the underlying infection dynamics, which are then linked to the number of reported cases via time-varying reporting probabilities.

We have extensively discussed the difficulties models encountered at turning points. In the aftermath of such events, epidemic forecasts typically receive increased attention in the general media (e.g., following the rapid downward turn in cases in Germany in May 2021[45]). While important from a subject-matter perspective, this is not without problems from a formal forecast evaluation standpoint. Major turning points are rare events and, as such difficult to forecast. Focusing evaluation on solely these instances will benefit models with a strong tendency to predict change, and adapting scoring rules to emphasize these events in a principled way is not straightforward. This problem is known as the forecaster's dilemma[46] in the literature and likewise occurs in, e.g., economics and meteorology (see illustrations in Table 1 from ref. [46]). An interesting question for future work is whether turning points are preceded by stronger disagreement between models, which might then serve as an alert; or whether, on the contrary, trend changes are followed by increased disagreement. Especially the latter question has received considerable attention in economic forecasting[47].

In this paper, we only applied unweighted ensembles and a heuristic, rather unflexible weighting scheme based directly on the past average performance. More sophisticated weighting schemes have been explored in refs. [29] and[30] using data from the US COVID-19 Forecast Hub. Their results indicate that when some contributing forecasters have a stable record of good performance, giving these more weights can result in improved performance. In particular, restricting the ensemble to a set of well-performing models may be beneficial, a strategy employed in the so-called relative WIS weighted median ensemble[30] used by the US COVID-19 Forecast Hub since November 2021.

The present paper marks the end of the German and Polish COVID-19 Forecast Hub as an independently run platform. In April 2021, the European Center for Disease Prevention and Control (ECDC) announced the launch of a European COVID-19 Forecast Hub[4], which has since attracted submissions from more than 30 independent teams. The German and Polish COVID-19 Forecast Hub has been synchronized with this larger effort, meaning that all forecasts submitted to our platform are forwarded to the European repository, while forecasts submitted there are mirrored in our dashboard. In addition, we still collect regional-level forecasts, which are not currently covered in the European Forecast Hub. The adoption of the Forecast Hub concept by ECDC underscores the potential of collaborative forecasting systems with combined ensemble predictions as a key output, along with continuous monitoring of forecast performance. We anticipate that this closer link to public health policymaking will enhance the usefulness of this system to decision makers. An important step will be the inclusion of hospitalization forecasts. Due to unclear data access, these had not been tackled in the framework of the German and Polish COVID-19 Forecast Hub, but have been added in the new European version.

## Data availability
The forecast data generated in this study have been deposited in a GitHub repository (https://github.com/KITmetricslab/covid19-forecast-hub-de), with a stable Zenodo release available under accession code 5608390 https://zenodo.org/record/5608390#.YYFxdJso9H4. This repository also contains all case and death data used for evaluation. These have been taken from public sources of routine surveillance data[12–14], from which they can likewise be obtained. Forecasts can be visualized interactively at https://kitmetricslab.github.io/forecasthub/. An additional dataset summarizing all data shown in Figs. 1–7 has been made available in the Supplementary Material of this paper as Supplementary Data 1. Should any further data be required to reproduce the results, these can be obtained from the corresponding authors upon reasonable request.

## Code availability
Codes to reproduce figures and tables are available at https://github.com/KITmetricslab/analyses_de_pl2, with a stable version at https://zenodo.org/record/5639514#.YYF1aZso9H4[48]. The results presented in this paper have been generated using the release preprint2 of the repository https://github.com/KITmetricslab/covid19-forecast-hub-de, see above for the link to the stable Zenodo release. The codes require the R packages colorspace (version 2.0-3)[49], plotrix (version 3.8-1)[50], xtable (version 1.8-4)[51], and zoo (version 1.8-9[52]).

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

## Acknowledgements

J. Bracher, M. Schienle, and T. Gneiting acknowledge support from the Helmholtz Foundation via the SIMCARD Information and Data Science Pilot Project. J.Bracher and M.Schienle were moreover supported by the German Ministry of Research and Education via the project RESPINOW. T.Gneiting and D.Wolffram are grateful for support from the Klaus Tschira Foundation. D.Wolffram's. contribution was moreover supported by the Helmholtz Association under the joint research school HIDSS4Health— Helmholtz Information and Data Science School for Health as well as the German Federal Ministry of Education and Research (BMBF) and the Baden-Württemberg Ministry of Science as part of the Excellence Strategy of the German Federal and State Governments. N.I.Bosse was supported by the Health Protection Research Unit (grant code NIHR200908). S. Funk and S. Abbott were supported by the Wellcome Trust (210758/Z/18/Z). The itwm-dSEIR forecasting team (J. Fiedler, N. Leithäuser, and J. Mohring) was supported by the Ministry of Health and Science of Rhineland Palatinate and the Fraunhofer Anti-Corona Program. The LANL-GrowthRate forecasting team (L. Castro, G. Fairchild, and I.J Michaud) was supported by the Laboratory Directed Research and Development program of Los Alamos National Laboratory under project number 20200700ER. A. Srivastava was supported by National Science Foundation Awards 2027007 and 2135784. S. Bhatia acknowledges funding from the Wellcome Trust (219415). Work on the ICM UW epidemiological model (J.M. Nowosielski, M. Radwan, and F. Rakowski) was supported by the Polish Minister of Science and Higher Education grant 51/WFSN/2020 given to the University of Warsaw. Development of the IMISE-SECIR model (Y. Kheifetz, H. Kirsten, and M. Scholz) was funded in the framework of the project SaxoCOV (Saxonian COVID-19 Research Consortium). SaxoCOV was co-financed with tax funds on the basis of the budget passed by the Saxon state parliament. The model presentation was funded by the NFDI4Health Task Force COVID-19 (www.nfdi4health.de/task-force-covid-19-2) within DFG project LO-342/17-1. Furthermore, the modeling of this group is funded by the Federal Ministry of Education and Research Germany (BMBF) within project PROGNOSIS (FKZ 031L0296A). The presented results build upon broader modeling efforts conducted by the different teams. We would like to acknowledge the complete teams for their contributions to these efforts: CovidAnalytics-DELPHI: Michael Lingzhi Li (Operations Research Center, Massachusetts Institute of Technology, Cambridge, MA, USA), Dimitris Bertsimas, Saksham Soni (both Sloan School of Management, Massachusetts Institute of Technology, Cambridge, USA). epiforecasts-EpiExpert and epiforecasts-EpiNow2: Sam Abbott, Nikos I. Bosse, Sebastian Funk (all London School of Hygiene and Tropical Medicine, London, UK). FIAS_FZJ-Epi1Ger: Maria Vittoria Barbarossa (Frankfurt Institute for Advanced Studies, Frankfurt, Germany), Jan Fuhrmann (Institute of Applied Mathematics, University of Heidelberg, Heidelberg, Germany), Jan H. Meinke (Jülich Supercomputing Centre, Forschungszentrum Jülich, Jülich, Germany). SDSC_ISG-TrendModel: Antoine Flahault, Elisa Manetti, and Kristen Namigai (all Institute of Global Health, Faculty of Medicine, University of Geneva, Geneva, Switzerland), Christine Choirat, Benjamin Bejar Haro, Ekaterina Krymova, Gavin Lee, Guillaume Obozinski, and Tao Sun (all Swiss Data Science Center, ETH Zurich, and EPFL Lausanne, Switzerland), and Dorina Thanou (Center for Intelligent Systems, EPFL, Lausanne Switzerland). ICM-agentModel: Filip Dreger, Łukasz Górski, Magdalena Gruziel-Słomka, Artur Kaczorek, Antoni Moszyński, Karol Niedzielewski, Jedrzej Nowosielski, Maciej Radwan, Franciszek Rakowski, Marcin Semeniuk, and Jakub Zieliński (all Interdisciplinary Centre for Mathematical and Computational Modeling, University of Warsaw, Warsaw, Poland), Rafał Bartczuk (Interdisciplinary Centre for Mathematical and Computational Modeling, University of

Warsaw, Warsaw and Institute of Psychology, John Paul II Catholic University of Lublin, Lublin, Poland), Jan Kisielewski (Interdisciplinary Centre for Mathematical and Computational Modeling, University of Warsaw, Warsaw and Faculty of Physics, University of Białystok). Imperial-ensemble2: Sangeeta Bhatia (MRC Centre for Global Infectious Disease Analysis, Abdul Latif Jameel Institute for Disease and Emergency Analytics (J-IDEA), Imperial College, London, UK), Pierre Nouvellet (School of Life Sciences, University of Sussex, Brighton, UK). itwm-dSEIR: Michael Burger, Robert Feßler, Jochen Fiedler, Michael Helmling, Karl-Heinz Küfer, Neele Leithäuser, Jan Mohring, Johanna Schneider, Anita Schöbel, Michael Speckert, Raimund Wegener, and Jarosław Wlazło (all Fraunhofer Institute for Industrial Mathematics, Kaiserslautern, Germany). ITWW-county_repro: Przemyslaw Biecek (Warsaw University of Technology, Warsaw, Poland), Viktor Bezborodov, Marcin Bodych, and Tyll Krueger (all Wroclaw University of Science and Technology, Poland), Jan Pablo Burgard (Economic and Social Statistics Department, University of Trier, Germany), Stefan Heyder and Thomas Hotz (both Institute of Mathematics, Technische Universität Ilmenau, Ilmenau, Germany) LANL-GrowthRate: Dave A. Osthus and Isaac J. Michaud (both Statistical Sciences Group, Los Alamos National Laboratory, Los Alamos, USA), Lauren Castro and Geoffrey Fairchild (both Information Systems and Modeling, Los Alamos National Laboratory, Los Alamos, USA). LeipzigIMISE-SECIR: Yuri Kheifetz, Holger Kirsten, and Markus Scholz (all Institute for Medical Informatics, Statistics and Epidemiology, University of Leipzig, Leipzig, Germany). MIMUW-StochSEIR: Anna Gambin, Krzysztof Gogolewski, Błażej Miasojedow, and Ewa Szczurek (all Institute of Informatics, University of Warsaw, Warsaw, Poland), Daniel Rabczenko and Magdalena Rosińska (Polish National Institute of Public Health—National Institute of Hygiene). MOCOS-agent1: Marek Bawiec, Viktor Bezborodov, Marcin Bodych, Radosław Idzikowski, Tyll Krueger, Tomasz Ożański, Ewaryst Rafajłłowicz, Ewa Skubalska-Rafajłowicz, and Wojciech Rafajłowicz (all Wroclaw University of Science and Technology, Poland), Barbara Pabjan (Institute of Sociology University of Wroclaw, Poland,), Przemyslaw Biecek (Warsaw University of Technology), Agata Migalska (Wroclaw University of Science and Technology, Poland and Nokia Solutions and Networks, Wroclaw, Poland), and Ewa Szczurek (University of Warsaw). USC-SIkJalpha: Ajitesh Srivastava and Frost Tianjian Xu (both University of Southern California, Los Angeles, USA). We moreover thank Dean Karlen for contributing forecasts and the Institute for Health Metrics and Evaluation, University of Washington, for making forecasts publicly available under a free license. We are moreover grateful for the support and advice from the organizing team of the US COVID-19 Forecast Hub. The content of this manuscript is solely the responsibility of the authors and does not necessarily represent the official views of the institutions they are affiliated with.

## Author contributions

J.B., D.W., T.G., and M.Se. conceived the study with advice from A.U. J.B., D.W., J.D., K.G., and J.L.K. put in place and maintained the forecast submission and processing system. A.U. coordinated the creation of an interactive visualization tool. J.B. performed the evaluation analyses with inputs from D.W., T.G., M.Se., and members of various teams. S.A., M.V.B., D.B., S.B., M.B., N.I.B., J.P.B., L.C., G.F., J.Fr., J.Fn., S.F., A.G., K.G., S.H., T.H., Y.K., H.K., T.K., E.K., M.L.L., J.H.M., B.M., I.J.M., J.H.M., J.Mg., P.N., J.M.N., T.O., M.R., F.R., M.Sz., S.S., and A.S. contributed forecasts (see list of contributors by the team). J.B., T.G., and M.Se. wrote the manuscript. All teams and members of the coordinating team provided feedback on the manuscript and descriptions of the respective models.

## Funding

## Competing interests

The authors declare no competing interests.

## Additional information

Johannes Bracher [1,2✉], Daniel Wolffram [1,2,3], Jannik Deuschel [1], Konstantin Görgen [1], Jakob L. Ketterer [1], Alexander Ullrich [4], Sam Abbott [5], Maria V. Barbarossa [6], Dimitris Bertsimas [7], Sangeeta Bhatia [8], Marcin Bodych [9], Nikos I. Bosse [5], Jan Pablo Burgard [10], Lauren Castro [11], Geoffrey Fairchild [11], Jochen Fiedler [12], Jan Fuhrmann [13], Sebastian Funk [5], Anna Gambin [14], Krzysztof Gogolewski [14], Stefan Heyder [15], Thomas Hotz [15], Yuri Kheifetz [16], Holger Kirsten [16], Tyll Krueger [9], Ekaterina Krymova [17], Neele Leithäuser [12], Michael L. Li [18], Jan H. Meinke [19], Błażej Miasojedow [14], Isaac J. Michaud [20], Jan Mohring [12], Pierre Nouvellet [21], Jedrzej M. Nowosielski [22], Tomasz Ozanski [9], Maciej Radwan [22], Franciszek Rakowski [22], Markus Scholz [16], Saksham Soni [18], Ajitesh Srivastava [23], Tilmann Gneiting [2,24] & Melanie Schienle [1,2✉]

[1]Chair of Statistical Methods and Econometrics, Karlsruhe Institute of Technology (KIT), Karlsruhe, Germany. [2]Computational Statistics Group, Heidelberg Institute for Theoretical Studies (HITS), Heidelberg, Germany. [3]HIDSS4Health - Helmholtz Information and Data Science School for Health, Karlsruhe/Heidelberg, Germany. [4]Robert Koch Institute (RKI), Berlin, Germany. [5]London School of Hygiene and Tropical Medicine, London, UK. [6]Frankfurt Institute for Advanced Studies, Frankfurt, Germany. [7]Sloan School of Management, Massachusetts Institute of Technology, Cambridge, MA, USA. [8]MRC Centre for Global Infectious Disease Analysis, Abdul Latif Jameel Institute for Disease and Emergency Analytics (J-IDEA), Imperial College London, London, UK. [9]Wroclaw University of Science and Technology, Wroclaw, Poland. [10]Economic and Social Statistics Department, University of Trier, Trier, Germany. [11]Information Systems and Modeling, Los Alamos National Laboratory, Los Alamos, NM, USA. [12]Fraunhofer Institute for Industrial Mathematics (ITWM), Kaiserslautern, Germany. [13]Institute for Applied Mathematics, University of Heidelberg, Heidelberg, Germany. [14]Faculty of Mathematics, Informatics, and Mechanics, University of Warsaw, Warsaw, Poland. [15]Institute of Mathematics, Technische Universität Ilmenau, Ilmenau, Germany. [16]Institute for Medical Informatics, Statistics and Epidemiology, University of Leipzig, Leipzig, Germany. [17]Swiss Data Science Center, ETH Zürich and EPF Lausanne, Zürich, Switzerland. [18]Operations Research Center, Massachusetts Institute of Technology, Cambridge, MA, USA. [19]Jülich Supercomputing Centre, Forschungszentrum Jülich, Jülich, Germany. [20]Statistical Sciences Group, Los Alamos National Laboratory, Los Alamos, NM, USA. [21]School of Life Sciences, University of Sussex, Brighton, UK. [22]Interdisciplinary Centre for Mathematical and Computational Modelling, University of Warsaw, Warsaw, Poland. [23]Ming Hsieh Department of Computer and Electrical Engineering, University of Southern California, Los Angeles, CA, USA. [24]Institute for Stochastics, Karlsruhe Institute of Technology (KIT), Karlsruhe, Germany. ✉email: johannes.bracher@kit.edu; melanie.schienle@kit.edu

