## [Peer Review File · Communications Medicine]

Reviewers' comments:

Reviewer #1 (Remarks to the Author):

Bracher et al. provide a straightforward, interesting, and timely paper that discusses the possibilities and challenges associated with probabilistic forecasting in a pandemic that has been dictated by extreme inflection points. Generally, this paper was easy to read and interpret with nice visualizations.

Major comments

-Lines 227-228: is it possible that meaningful case forecasts are only possible at short horizons because of reporting problems? Do you think the same would be the case if you were forecasting infections? Or cases where you accounted (scaled) by some reporting difference? This could be interesting to explore (not necessarily in this manuscript, but more generally). Either way, can you please discuss this.

-Have you considered a forecasting metric that evaluates the similarity (or difference) among forecasts? It would be interesting to see how some kind of similarity index changes through time. I.e., maybe at points in time where things are trending up or down, forecasts from different models more or less converge, but at inflection points or with new variants arising, there could be deviation among different models. More philosophically this could be point to the importance an ensemble of models. It would be useful to explore some kind of metric like this.

-Given the readership of Communications Medicine is not necessarily forecasting-focused, I think it could be useful to provide some more intuitive definitions for some of the different forecasting terminology. It would also be useful to describe in some of the figure captions how to interpret certain figure types (e.g., the empirical coverage plots).

Minor comments

-Line 29-30: If you're going to include this, it might be nice to give an explicit example as to what the "need" is... i.e., does the German-speaking public discourse not trust a study unless it is pre-registered?

-Line 52-54: did you consider adjusting the government stringency index to account for changes in daycare or other changes that were not reflected in the index?

-Line 98-99: why would it be different for deaths versus cases? The trends in death counts (fig 1) actually seems to be more extreme than cases?

-Line 125-127: have you considered normalizing the WIS for comparison purposes? Is this the average weighted interval score? Can you clarify this?

-Lines 121-123: why was it different with case forecasts in Germany? What is the intuition?

- Line 108-110 and elsewhere: I'm not sure how well known the "undercoverage" and "overcoverage" terminology are for a medical journal. It might be useful to describe this more.

-Figures 2-3: I think the empirical coverage plots are useful, but they are quite novel in terms of viewing forecast performance (although I know they are also presented in the Nat Comm paper). I would just recommend giving an intuition to how to view those plots (i.e., what makes one model "good" in this respect versus another model), either in the results or in the caption.

-Figure 4: The absolute error for the 4-week horizon are outside of the plotting area. Please fix this.

-Figure 5: It might be useful to note in the caption *which* two models accounted for both the establishment of another variant (and when those changes were made to the models). It would be easier to see how those models compared to the other models that did not account for this.

-Figure 7: It is far too difficult to see the coloring of the yellow line, can you please adjust this color scheme to ensure all lines are visible?

Reviewer #2 (Remarks to the Author):

The manuscript focuses on the evaluation of forecasts for Covid-19 pandemic in Germany and Poland. In general, the content is highly valuable to the community, but the manuscript needs some improvements before acceptance. Below are my detailed comments.

- Some of the key references are not included. For example, <https://www.nature.com/articles/s41746-021-00511-7> presents a similar study for US and Japan.
- Fig. 4 is too complex as it is trying to show many quantities in one figure. I suggest simplifying this figure and putting the unnecessary content into Appendix.
- Testing rates have evolved in both countries, and lack of testing would affect the ground truth case counts and hence the credibility of evaluation metrics. How do the authors take into account in discussions?
- Section 2.2 is highly valuable, but there should be more discussions on the takeaways. For examples, what are the most important modeling considerations to not miss the changes in the trends or peaks? How can other variables like NPIs help with that? Which models are particularly strong in different regimes?
- In Eq. (2), rather than hardcoding the number 11, the more general form can be presented.
- Ensembling methods are highly heuristic - there is no justification on why specific forms are used.
- Table 3 does not include some other key modeling aspects like vaccination and mobility.
- Results for 4-weeks ahead are the most important for resource planning, but only shown for a subset of tables/figures.

Reviewer #3 (Remarks to the Author):

This manuscript describes the evaluation of real-time forecasts of COVID-19 cases and deaths in Germany and Poland. Forecasts were made by different teams and combined into an ensemble model.

Overall, the manuscript is well written and is particularly relevant in the context of the on-going pandemic of COVID-19.

Although the approach is not new, and has already been described in a previous paper from the authors (Bracher et al, Nature Communications 2021) (the present study is a follow-up study, conducted on a more recent time period), I find it useful to see a comparison of the models over two time periods, with different characteristics and different dynamics.

The authors manage to summarize the numerous results of the evaluation in a comprehensible way. In addition, they choose to discuss into more details a particular aspect of their results, i.e. the performance of the models at inflection points, which I find particularly interesting.

Their analysis should be of great interest for other researchers working on short-term forecasting (of any disease).

I only have minor comments:

- Page 2, line 68 : "seven contributed models" should read "five contributed models"
- Figure 4: why aren't all models displayed?
- Legend of figure 4: "The grey area represents the performance of the baseline model KIT-baseline." Can you clarify? WIS of baseline = bottom of the grey area? What is the grey line? AE ?

- All tables: Why giving coverage as fractions? I think decimal numbers would be much easier to compare to the nominal coverage.

Reviewer #4 (Remarks to the Author):

The manuscript addresses an interesting and timely topic. The ensemble approach is very useful and is a practical and effective alternative to single-based forecasts. The proposed approach can be used to predict/forecast the evolution of the epidemic and may be a very useful tool to manage the epidemic, under uncertainty.

1. I would suggest to add more details on how the weights of the ensemble approach are obtained/estimated. Please, comment on the potential impact of missing data imputation method employed on the forecasts. Moreover, a reference to the Richar
2. I am wondering if a sort of selection procedure can be implemented if any of the considered models would provide unexpected unreliable estimates. Of course, the weights will play a crucial role to avoid that a model with unreliable estimates/forecasts drives the entire ensemble estimate. At the same time, however, as one-model-fit-all is not a reasonable option, it would be nice to see under which conditions the ensemble approach depends more on: agent-based, compartmental, growth models, etc. In other words, it would be interesting if a category of models should be in general preferred, or under which conditions a specific category performs better than others.
3. It is in general difficult to predict the peak and, after that, the evolution of the epidemic. Some of the considered models require some time to adjust if some conditions change. I am wondering if the uncertainty surrounding the estimates at the peak, inflection points or when a new variant arises, is larger than "usual". This is because even looking at the uncertainty surrounding the estimates may be an indication, an alert of something changing.
4. At last, you discuss short-term forecasts, with a focus on 1 and 2 weeks ahead forecasts. Does these forecasts depend on the length of training data considered? Could you define an optimal length or is it always better to consider the entire data from a wave?

COMMSMED-21-0564-T: Point-by-point response to the referees' comments

Johannes Bracher, Daniel Wolffram, Jannik Deuschel,
Konstantin Gorgen, Jakob Ketterer et al.

29th June 2022

We would like to thank the review team for the constructive reports and helpful comments on our paper. In what follows we respond to the different points raised by the reviewers. The page references in the referees' statements refer to the original version. We also provide a version of the revised paper with track changes.

Remarks on reporting completeness and testing policies

As both Referee 1 (comment 1) and Referee 2 (comment 3) have raised the question of reporting completeness / changing testing strategies, we provide some general information here to which we will refer in the replies to both referees.

Case incidence rates are subject to under-ascertainment, the degree of which depends on testing policies and capacities. Death counts, too, may be underreported, though usually to a lesser degree (see the discussion of discrepancies between death reports and general excess mortality in Poland in our previous manuscript Bracher et al 2021, reference [13]).

A relevant change in both the German and Polish testing strategy during the considered period was the increased use of rapid lateral flow tests (LFTs) starting in March 2021. From 8 March 2021 onward, citizens in Germany had access to free rapid tests in local testing centers. In Poland a new online system to provide access to tests without the need for a prescription by a doctor was announced on 15 March 2021. By enabling targeted testing of persons with positive LFTs, this likely had an impact on the composition of the group of persons receiving PCR tests.

As can be seen from the middle row of Figure R1, the PCR testing volume did not change drastically in Germany during the considered period (marked in dark grey), while in Poland it considerably increased. The percentage of positive tests also considerably increased to up to 33% in Poland from late February onwards, while in Germany it

Figure R1: Top row: Seven-day sums of newly reported cases of COVID-19 in Germany and Poland, Nov 2020 – April 2021. Middle row: Number of PCR tests performed per 1,000 inhabitants in Germany and Poland. Bottom row: Test positivity percentages over the same period. Both quantities are shown as seven-day rolling averages. The light and dark grey rectangles highlight the study periods of our previous study [13] and the current manuscript, respectively. Data: RKI / Polish Ministry of Health, compiled by *Our World in Data*.

followed a U-shape (similar to the observed case incidence) at a lower level. While these data do not allow for clear-cut conclusions on the evolution of the case detection probability, it seems likely that the latter varied over time. In particular, the very high test positivity rate in Poland at the end of our study period may be indicative of an overload of the testing system, resulting in lower case detection probabilities. Note that the spike in the number of tests performed and the positivity percentage visible in Poland in January is likely an artefact due to a data dump rather than a genuine feature of the data.

We have added a panel showing the proportion of positive tests to Figure 1 to direct the reader’s attention to this aspect, along with the following explanation:

Panel (d) shows the proportion of all performed PCR tests which turned out positive. While in Germany the curve follows a U-shape similar to the case incidence curve, the test positivity rate continuously increased in Poland,

peaking at 33%.

Moreover we added the following paragraph to the discussion section:

Another difficulty of case forecasts is incomplete case ascertainment, which must be assumed to vary over time (see e.g., the discussions in Arik et al. 2021 and Fuhrmann and Barbarossa 2020). As a consequence, data can be difficult to compare across different phases of the pandemic, and modellers often choose to only use a recent subset of the available data to calibrate their models. While data on testing volumes and test positivity rates are available, estimation of the reporting fractions and anticipation of its future development is challenging. Even if models correctly reflect underlying epidemic dynamics, this may thus not translate to accurate forecasts of the observed number of confirmed cases. This is a limitation of the considered forecasts and their evaluation, which inherit the difficulties of the underlying truth data sources. Nonetheless, we argue that a distinguishing feature of forecasts is that they refer to observable quantities, and forecasters should take into account all relevant aspects of the system producing them. Indeed, many of the considered models (e.g., MOCOS-agent1 and FIAS_FZJ-Epi1Ger) attempt to reconstruct the underlying infection dynamics, which are then linked to the number of reported cases via time-varying reporting probabilities.

Response to Referee 1

Bracher et al. provide a straightforward, interesting, and timely paper that discusses the possibilities and challenges associated with probabilistic forecasting in a pandemic that has been dictated by extreme inflection points. Generally, this paper was easy to read and interpret with nice visualizations.

Major comments:

1: *Lines 227-228: is it possible that meaningful case forecasts are only possible at short horizons because of reporting problems? Do you think the same would be the case if you were forecasting infections? Or cases where you accounted (scaled) by some reporting difference? This could be interesting to explore (not necessarily in this manuscript, but more generally). Either way, can you please discuss this.*

Reply: We agree that variations in testing policies and the resulting reporting completeness (see discussion above) represent an additional layer of complexity in the considered forecasting task. We assume that these changes usually occur gradually over time and thus agree with the referee that they are of greater importance for longer horizons (as larger discrepancies can build up over time).

The idea of predicting infections rather than reported cases is an interesting one. Indeed many of the considered models also attempt to reconstruct the total numbers of infections, extrapolate their dynamics and then translate this to the observed case

Figure R2: Reconstruction of infection dynamics by the FIAS_FZJ model.

numbers via a reporting process. As an example, Figure R2 shows the dynamics of infections as reconstructed by the FIAS_FZJ model, with pronounced temporal variability in reporting completeness (ratio between the red and yellow/orange areas).

A difficulty when predicting actual infections rather than reported infections is that the former remain an unobserved quantity even in hindsight. The Forecast Hub concept, on the other hand, is based on the principle of predicting observable quantities, thus enabling evaluation of forecasts against later observed data. This, of course, is not feasible for the total number of infections. We therefore focus on predicting observed cases, even if it means forecasters also need to take into account the reporting process.

We added a paragraph on this aspect (along with comment 3 from Referee 2) in the Discussion section, see Section “Remarks on reporting completeness and testing policies” above.

2: *Have you considered a forecasting metric that evaluates the similarity (or difference) among forecasts? It would be interesting to see how some kind of similarity index changes through time. I.e., maybe at points in time where things are trending up or down, forecasts from different models more or less converge, but at inflection points or with new variants arising, there could be deviation among different models. More philosophically this could be point to the importance an ensemble of models. It would be useful to explore some kind of metric like this.*

Reply: This is a very interesting thought. Our reply to this suggestion is twofold:

- We are indeed working on an analysis along these lines in collaboration with our colleagues from the US COVID-19 Forecast Hub. Specifically we are using the

Cramér distance to this end. The analyses are focused on two aspects: firstly, as suggested by the reviewer, we are assessing in how far disagreement between different models is an indicator for changing trends. Secondly, we are assessing whether forecasts based on similar modelling approaches (e.g., all forecasts from compartmental models), are more similar than forecasts based on very different approaches. To this end we are currently developing a quantile-based approximation of the Cramér distance, which follows the same general idea as the approximation of the CRPS by the weighted interval score (WIS).

- For the present work we are unfortunately unable to apply the approach outlined in the previous point. The main reasons are that the underlying theoretical development (quantile-based approximation of the Cramér distance) is not completed and that the number of available forecasts in the German and Polish project is relatively small. In order to compare the similarity of forecasts over time one needs to select a set of models which are available for every single time point. Unfortunately, many models in our collaboration were not submitted every single week of our study period (in particular during the first part before Christmas 2020) or did not provide complete sets of quantiles. We are thus left either with a quite small set of models or a small set of weeks to consider.

We therefore attempted a simpler approach for the forecasts presented in this paper where we only assess the similarity of point forecasts of cases. For Germany the models with complete sets of point forecasts during the two study periods are `epiforecasts-EpiExpert`, `epiforecasts-EpiNow2`, `FIAS_FZJ-Epi1Ger`, `ITWW-county_repro`, `LANL-GrowthRate`, `LeipzigIMISE-SECIR`, and `USC-SIkJalpha`. For Poland, `epiforecasts-EpiExpert`, `epiforecasts-EpiNow2`, `ITWW-county_repro`, `LANL-GrowthRate`, `MOCOS-agent1`, and `USC-SIkJalpha` could be used. Figure R3 shows the time series of observations from Figure 1 (top row) along with the standard deviations of one-week ahead point forecasts for cases made at different time points. In the second row we show the standard deviation of point forecasts on their original scale. It can be seen that it scales with the level of magnitude of the observed counts. This is to be expected as the same differences in expected growth rate will translate to larger standard deviations on the absolute scale. For this reason we also show standard deviations of log-transformed point forecasts (bottom row), which are assumed to be less scale-dependent and thus more suitable for our purpose.

In our view the resulting graphs do not show any easily interpretable patterns. For Poland the highest degree of heterogeneity (of log-transformed point forecasts) occurred during a phase of continued upward trend and was due to a single somewhat unusual forecast which differed considerably from the rest. For Germany, the largest heterogeneity was observed in November following first signs of a trend change towards a plateau. Some models picked up on this change, while others did not. In this case it thus seems like the most pronounced heterogeneity occurred right after the first signs of change.

Figure R3: Observed time series of weekly cases (top row) and standard deviation of point forecasts from different models (see list in text). We show both standard deviations of point forecasts on the original scale and log-transformed point forecasts, which are assumed to be less scale-dependent.

Due to this somewhat inconclusive overall picture and the fact that a similar study in its own right is currently in preparation we decided not to include the created figure into the manuscript or Supplementary Information. Instead, we treat this aspect as an open question in the discussion section:

An interesting question for future work is whether turning points are preceded by stronger disagreement between models, in which case this might serve as an alert; or whether, on the contrary, trend changes are followed by increased disagreement. Especially the latter question has received considerable attention in economic forecasting (Coibion et al 2012).

3: *Given the readership of Communications Medicine is not necessarily forecasting-focused, I think it could be useful to provide some more intuitive definitions for some of the different forecasting terminology. It would also be useful to describe in some of the figure captions how to interpret certain figure types (e.g., the empirical coverage plots).*

Reply: We have added a description on how to interpret the coverage plots to Figure 2. Moreover we made the following edits and additions to make forecast-related terminology more accessible:

- page 1: we explained the term “ensemble forecast” as “combinations of different available forecasts”.
- page 2: we explain “well-calibrated” as “prediction intervals contained the true value roughly as often as intended”.
- page 2: we explain “hindsight bias” as “the tendency to overstate the predictability of past events in hindsight”.
- page 3: changed “naïve last-observation-carried-forward model” to “a naïve model always using the last observed value as the expectation for the following weeks.”
- page 3: we explain “targets” as “quantities to be predicted”.
- page 9: we clarified that “inflection points” are just “changes in trend”.

Minor comments:

4: *Line 29-30: If you’re going to include this, it might be nice to give an explicit example as to what the “need” is... i.e., does the German-speaking public discourse not trust a study unless it is pre-registered?*

Reply: We made this statement more explicit and now state that the purpose of pre-registrations (as pointed out in reference Dirnagel 2021) is to avoid hindsight bias. Hindsight bias is a common cognitive bias and describes the tendency to perceive past events as having been more predictable than they actually were (often in conjunction with the claim that a given model could have predicted an event had it been used in real time).

In the context of predictive epidemic modelling it can either occur if forecasts are not actually generated in real time, but only retrospectively by applying models to old data (so-called *hindcasts*) or if the evaluation strategy is adjusted to the observed performance (e.g., selective reporting excluding time points when forecasts failed). Our prospective and pre-registered study avoids both pitfalls.

The respective paragraph now reads as follows:

Also, it forms the basis for a systematic evaluation of performance. This is a prerequisite for model consolidation and improvement, and a need repeatedly expressed (Nature Publishing Group 2020). It has been highlighted that such modelling studies should be prospective (Arik et al 2021) and ideally follow pre-registered protocols (Dirnagel 2021) in order to prevent selective reporting and hindsight bias (i.e., the tendency to overstate the predictability of past events in hindsight).

5: *Line 52-54: did you considering adjusting the government stringency index to account for changes in daycare or other changes that were not reflected in the index?*

Reply: Within the group of authors we had some discussions on how to display the level of non-pharmaceutical interventions. In the end we reached a consensus that despite a slight discrepancy with our perception of the new restrictions in Poland in March 2021, the Oxford Stringency Index was a helpful summary measure. We decided to add a remark on these restrictions in the verbal description, but did not consider explicitly adjusting the index. Given the elaborateness of the index a thoroughly justified adjustment did not seem feasible to us.

6: *Line 98-99: why would it be different for deaths versus cases? The trends in death counts (fig 1) actually seems to be more extreme than cases?*

Reply: It is correct that deaths, too, overall follow a relatively clear trend over the study period. The extrapolation baseline model thus provides rather reasonable predictions. However, deaths are in general easier to predict as they are a more lagged indicator of the pandemic dynamics (see Discussion section). As models were able to exploit information from earlier indicators like cases and hospitalizations, they were to some degree able to predict the deviations from a pure trend continuing pattern. For cases this seems to be more challenging.

To make this more clear we have re-phrased the respective passage as follows:

Given the relatively long stretches of continued upward or downward trends in cases, this simple heuristic was not easy to beat and is rather close to the performance of the ensemble forecasts. For deaths, too, there are rather clear trends over the study period. Nonetheless, the different ensemble forecasts achieve substantial improvements over `KIT-extrapolation_baseline`, meaning that the deviations from the overall trends were predicted with some success.

7: *Line 125-127: have you considered normalizing the WIS for comparison purposes? Is this the average weighted interval score? Can you clarify this?*

Reply: Yes, this is indeed the average WIS. We have made sure all tables and figures showing the WIS contain a clarification that it is the average weighted interval score. There is indeed a so-called “relative WIS”, which has been used e.g. in the referenced paper by Cramer et al (2022). It is defined as the ratio of the mean WIS of the model in question and the baseline model. Values below 1 and above 1 then indicate better or worse performance than the baseline, respectively. Unfortunately, space does not permit reporting both untransformed and standardized AE/WIS in the same table. We opted for the former as it has the following appealing interpretation: informally speaking, it can be seen as the mean absolute error after an adjustment for the included prediction uncertainty (see Section 4.2). It thus makes an interpretable statement about quality of forecasts on the natural scale of the data. Notably, it conveys how forecast performance declines over time, an aspect which would be lost when normalizing the score. To complement this, a relative version of the average scores is now available in the Supplementary Material (Table S1), with text reference as follows:

We here show the average scores on the absolute scale, where they can be interpreted as the average distance between the observed and predicted value (the WIS moreover taking into account forecast uncertainty). A summary table of relative scores standardized by the performance of the naïve `KIT-baseline` model is available in Supplementary Table S1.

8: *Lines 121-123: why was it different with case forecasts in Germany? What is the intuition?*

Reply: We have visually re-inspected the forecasts, but it is difficult to provide a convincing intuitive explanation. The general difficulty of predicting cases compared to deaths is certainly one factor, but beyond this all explanations are somewhat speculative. We thus refrained from providing such an explanation in the manuscript and restrict ourselves to the stated quantitative finding.

9: *Line 108-110 and elsewhere: I’m not sure how well known the “undercoverage” and “overcoverage” terminology are for a medical journal. It might be useful to describe this more.*

Reply: We have added an explanation of these terms at their first occurrence:

- “undercoverage (i.e., prediction intervals cover the observations less frequently than intended)”
- “overcoverage (intervals cover more often than intended)”

10: *Figures 2-3: I think the empirical coverage plots are useful, but they are quite novel in terms of viewing forecast performance (although I know they are also presented in the*

Nat Comm paper). I would just recommend giving an intuition to how to view those plots (i.e., what makes one model “good” in this respect versus another model), either in the results or in the caption.

Reply: Thank you for pointing out this missing explanation – the plots were indeed not understandable to a reader unfamiliar with this specific type of display. We have added an explanation to the legend of Figure 2:

“The dark and light bars represent the proportion of cases where the 50% and 95% prediction intervals, respectively, contained the observed values. Dotted lines show the desired nominal levels 0.5 and 0.95.”

11: Figure 4: The absolute error for the 4-week horizon are outside of the plotting area. Please fix this.

Reply: At the request of Reviewer 2 (comment 2) we simplified Figure 4 and removed the decomposition. This also allowed us to use a square-root scale to improve visibility and remedy the problem of points outside of the plot region.

12: Figure 5: It might be useful to note in the caption **which** two models accounted for both the establishment of another variant (and when those changes were made to the models). It would be easier to see how those models compared to the other models that did not account for this.

Reply: We now state that the only models explicitly accounting for the presence of multiple variants (starting on 1 March) are **Karlen-pypm** and **LeipzigIMISE-SECIR**.

13: Figure 7: It is far too difficult to see the coloring of the yellow line, can you please adjust this color scheme to ensure all lines are visible?

Reply: We agree with the referee. We have changed the colour scheme and now avoid using yellow lines.

Response to Referee 2

The manuscript focuses on the evaluation of forecasts for Covid-19 pandemic in Germany and Poland. In general, the content is highly valuable to the community, but the manuscript needs some improvements before acceptance. Below are my detailed comments.

1: Some of the key references are not included. For example, <https://www.nature.com/articles/s41746-021-00511-7> presents a similar study for US and Japan.

Reply: Thank you for this reference. We now reference this work to highlight the importance of evaluating models prospectively rather than retrospectively (Introduction section) and concerning the problem of incomplete case reporting (Discussion section).

2: Fig. 4 is too complex as it is trying to show many quantities in one figure. I suggest simplifying this figure and putting the unnecessary content into Appendix.

Reply: We simplified this figure by removing the decomposition of the WIS. We moreover moved to a square-root scale for the y-axis to enhance readability. The plot including the decomposition has been moved to the Supplement.

3: *Testing rates have evolved in both countries, and lack of testing would affect the ground truth case counts and hence the credibility of evaluation metrics. How do the authors take into account in discussions?*

Reply: We provide a short description of how testing policies and testing volume evolved over the considered period in a separate section of this response letter, see above. We agree with the referee that these aspects have a relevant effect on reported case numbers, which may not be directly comparable over time and countries. Forecasts referring to these time series inherit these limitations, as do the resulting model evaluations. We added a paragraph on this aspect to the discussion section, see also our reply to comment 1 from Reviewer 1:

Another difficulty of case forecasts is incomplete case ascertainment, which must be assumed to vary over time (see e.g., the discussions in Arik 2021 and Fuhrmann and Barbarossa 2020). As a consequence, data can be difficult to compare across different phases of the pandemic, and modellers often choose to only use a recent subset of the available data to calibrate their models. While data on testing volumes and test positivity rates are available, estimation of the reporting fractions and anticipation of its future development is challenging. Even if models correctly reflect underlying epidemic dynamics, this may thus not translate to accurate forecasts of the observed number of confirmed cases. This is a limitation of the considered forecasts and their evaluation, which inherit the difficulties of the underlying truth data sources. Nonetheless, we argue that a distinguishing feature of forecasts is that they refer to observable quantities, and forecasters should take into account all relevant aspects of the system producing them. Indeed, many of the considered models (e.g., MOCOS-agent1 and FIAS_FZJ-Epi1Ger) attempt to reconstruct the underlying infection dynamics, which are then linked to the number of reported cases via a reporting process.

4: *Section 2.2 is highly valuable, but there should be more discussions on the takeaways. For examples, what are the most important modeling considerations to not miss the changes in the trends or peaks? How can other variables like NPIs help with that? Which models are particularly strong in different regimes?*

Reply: We consider the following aspects crucial to enhance better predictions of trend changes:

- An improved understanding of the effect of NPIs. As more data on the impact of past NPIs become available, there is hope that their future effects can be anticipated better. The two Polish groups who correctly anticipated the turnaround in cases in Poland in March heavily based their forecasts on experience from previous sets of NPIs.

- Making use of the vastly improved availability of genetic surveillance data by extending models to multiple strains. Most models conceptually allow for such an extension, but in practice are often not adapted quickly enough.
- A better understanding of seasonal effects. The trend change in Germany in Spring 2021 (after our study period, but briefly discussed in Section 2.2) was likely partly driven by seasonality, but due to limited historical data this effect was difficult to quantify and anticipate at the time.
- A better understanding of population immunity in order to anticipate saturation effects (depletion of susceptibles). The increasing availability of seroprevalence data will be helpful in this respect, though the extension of models to include these may not be straightforward. Also, the emergence of new variants with (initially usually poorly understood) immune escape properties may pose limits to the practical benefits for short-term forecasting.

We have extended the paragraph on improved prediction of trend changes at the bottom of page 13 / top of page 14 in order to address these aspects in more detail:

“While the success of the two microsimulation models `MOCOS-agent1` and `ICM-agentModel` in anticipating the downward turn in cases in Poland remains a rather rare exception, it shows that careful mechanistic modelling combined with in-depth knowledge of national specificities has the potential to anticipate the impact of changing NPIs. As both groups heavily drew from experience from past NPIs in Poland, there is hope that predictions of the effects of NPIs will further improve as experience accumulates. An alternative strategy to improve case forecasts would be to identify appropriate leading indicators. These could for instance be trajectories in other countries (Harvey 2021) or additional data streams on e.g., mobility, insurance claims or web searches. However, the benefits of such data for short-term forecasting thus far have been found to be modest (McDonald et al 2021). Changes in dominant variants may make changes in overall trends predictable as they arise from the superposition of adverse but stable trends for the different variants. The availability of sequencing data has improved considerably since our study period, and we consider the extension of models to accommodate multiple strains a key step towards improved prediction of trend changes. Other relevant aspects include seasonal effects, which during our study period remained poorly understood due to limited historical data, and population immunity. As more data on seroprevalence become available, predictability of saturation effects may increase, though this will likely be complicated by the further evolution of the pathogen.”

5: *In Eq. (2), rather than hardcoding the number 11, the more general form can be presented.*

Reply: We have now adopted a more general display without hard-coding the number of used intervals.

6: *Ensembling methods are highly heuristic - there is no justification on why specific forms are used.*

Reply: We agree that the description of the ensemble approaches lacked contextualization and relevant references in the previous version of the manuscript. We have now added a paragraph linking the described ensemble approaches to the literature on the aggregation of quantile forecasts. Also, we state explicitly that the employed inverse-WIS approach is a heuristic approach which could be refined further:

Note that all forecast aggregations are performed directly at the level of quantiles rather than density functions as e.g., in Reich et al (2019). This approach is referred to as *Vincentization* (in reference to Vincent, 1912, see e.g., Busetti, 2017). A broader discussion of Vincentization approaches and their application to epidemiological forecasts, including numerous other weighting schemes, can be found in Taylor and Taylor (2021) and Ray et al (2022). Notably, Taylor and Taylor (2021) used a similar inverse score weighting approach and found it to perform well in a re-analysis of forecasts from the US COVID-19 Forecast Hub. In this context we note that our inverse-WIS ensemble does not involve any estimation or optimization of weights, but simply uses the inverse of an average of past scores as heuristic weights. A more flexible approach with one tuning parameter estimated from the data has been used in Ray et al (2022).

7: *Table 3 does not include some other key modeling aspects like vaccination and mobility.*

Reply: We agree that these aspects should be provided to the reader. Due to space reasons we were unable to include them into the table. As very few of the available models included used the mentioned additional data sources we opted for clarifying this aspect in the text instead:

During the evaluation period, only the `ICM-agentModel` explicitly accounted for vaccinations (given the low realized vaccination coverage by the end of the study period this aspect likely had limited impact). Only four models (`ICM-agentModel`, `Karlen-pypm`, `LeipzigIMISE-SECIR` and `MOCOS-agent1`, all only for certain weeks) explicitly accounted for the presence of multiple variants. In contrast to other related projects (Cramer et al 2021), none of the models used mobility or social media data.

8: *Results for 4-weeks ahead are the most important for resource planning, but only shown for a subset of tables/figures.*

Reply: We have re-discussed this question among the authorship group and have added a remark on the particular relevance of longer forecast horizons:

While we acknowledge the relevance of longer horizons for planning purposes, we argue that factors like changing non-pharmaceutical interventions and emergence of new variants limit meaningful forecasts (as opposed to scenarios) to rather short time horizons, especially for cases. As specified in our study protocol, we also report on the three and four week horizons, but defer parts of these analyses to the Supplementary Material.

In our discussions we reached the conclusion that as in the previous manuscript (Bracher et al 2021) we would like to keep a focus on shorter horizons. Especially for cases we concluded that the four-week horizon is outside the realm of forecasting, as essentially unpredictable features like changing NPIs or the emergence of new variants introduce too profound uncertainty. These should therefore be the subject of scenario modelling rather than forecasting. As in our study protocol we specified that the four-week horizon would be covered we provide the respective information in the Supplementary Material and selected figures, but we consider this shorter “boundary” of predictability a relevant takeaway from our project.

Response to Referee 3

This manuscript describes the evaluation of real-time forecasts of COVID-19 cases and deaths in Germany and Poland. Forecasts were made by different teams and combined into an ensemble model.

Overall, the manuscript is well written and is particularly relevant in the context of the on-going pandemic of COVID-19.

Although the approach is not new, and has already been described in a previous paper from the authors (Bracher et al, Nature Communications 2021) (the present study is a follow-up study, conducted on a more recent time period), I find it useful to see a comparison of the models over two time periods, with different characteristics and different dynamics.

The authors manage to summarize the numerous results of the evaluation in a comprehensible way. In addition, they choose to discuss into more details a particular aspect of their results, i.e. the performance of the models at inflection points, which I find particularly interesting.

Their analysis should be of great interest for other researchers working on short-term forecasting (of any disease).

I only have minor comments:

1: Page 2, line 68 : “seven contributed models” should read “five contributed models”

Reply: Thank you for spotting this. We have corrected this statement.

2: Figure 4: why aren't all models displayed?

Reply: Again thank you for spotting this. For space reasons we omitted the Imperial, IHME and SDSC models which only address the one-week-ahead horizon. We now state

this in the figure caption. The LeipzigIMISE-SECIR model, however, should have been in the figure, and we now added it (for Germany).

3: *Legend of figure 4: “The grey area represents the performance of the baseline model KIT-baseline.” Can you clarify? WIS of baseline = bottom of the grey area? What is the grey line? AE?*

Reply: Yes, this is indeed the AE. We now state the following in the figure legend:

The bottom end of the grey area represents the mean WIS of the baseline model KIT-baseline and the grey horizontal line its mean absolute error.

4: *All tables: Why giving coverage as fractions? I think decimal numbers would be much easier to compare to the nominal coverage.*

Reply: We have adapted this to decimal numbers and mention the number of evaluated forecasts in the figure caption instead.

Response to Referee 4

The manuscript addresses an interesting and timely topic. The ensemble approach is very useful and is a practical and effective alternative to single-based forecasts. The proposed approach can be used to predict/forecast the evolution of the epidemic and may be a very useful tool to manage the epidemic, under uncertainty.

1: *I would suggest to add more details on how the weights of the ensemble approach are obtained/estimated. Please, comment on the potential impact of missing data imputation method employed on the forecasts. Moreover, a reference to the Richar*

Reply: We now state explicitly that the weights of the inverse-WIS ensemble are not estimated, but based heuristically on an average of past scores. Also, we mention in Section 4.4 that missing submissions will lead to a penalization of the respective model and downweighting of its contribution to the inverse-WIS ensemble:

- (...)
- `KITCOVIDhub-inverse_wis_ensemble` The α -quantile of the ensemble forecast is a convex combination of the α -quantiles of the member forecasts. The weights are chosen inversely proportional to the mean WIS value obtained by the member models over the last six evaluated forecasts (last three one-week-ahead, last two two-week-ahead, last three-week-ahead). This is done separately for each time series to be predicted. Missing scores are imputed by the worst score achieved by any model for the respective target, meaning that irregularly submitted models will be penalized and receive less weight.

In the study protocol, the median ensemble was defined as our primary ensemble approach (Bracher et al 2020) as it can be assumed to be more robust

to occasional misguided forecasts (e.g., due to technical errors). We therefore display this version in all figures and focus our discussion on it. Note that all forecast aggregations are performed directly at the level of quantiles rather than density functions as e.g., in Reich et al (2019). This approach is referred to as *Vincentization* (in reference to Vincent, 1912, see e.g., Busetti, 2017). A broader discussion of Vincentization approaches and their application to epidemiological forecasts, including numerous other weighting schemes, can be found in Taylor and Taylor (2021) and Ray et al (2022). Notably, Taylor and Taylor (2021) used a similar inverse score weighting approach and found it to perform well in a re-analysis of forecasts from the US COVID-19 Forecast Hub. In this context we note that our inverse-WIS ensemble does not involve any estimation or optimization of weights, but simply uses the inverse of an average of past scores as heuristic weights. A more flexible approach with one tuning parameter estimated from the data has been used in Ray et al (2022).

Unfortunately this comment somehow got cut off and we were unable to identify the suggested reference. Please let us know via the review of the revised manuscript.

2: I am wondering if a sort of selection procedure can be implemented if any of the considered models would provide unexpected unreliable estimates. Of course, the weights will play a crucial role to avoid that a model with unreliable estimates/forecasts drives the entire ensemble estimate. At the same time, however, as one-model-fit-all is not a reasonable option, it would be nice to see under which conditions the ensemble approach depends more on: agent-based, compartmental, growth models, etc. In other words, it would be interesting if a category of models should be in general preferred, or under which conditions a specific category performs better than others.

Reply: A selection approach similar to the one suggested by the Referee is an interesting option. Indeed, such an approach has recently been implemented for the US COVID-19 Forecast Hub and is used e.g., in communications of the US CDC (see the newly referenced preprint by Ray et al 2022, co-authored by several authors of the present manuscript). As the manuscript by Ray et al is based on a considerably larger number of member forecasts and a longer time period, we have decided to refer to this work to cover the proposed aspect. We have added the following paragraph to the discussion:

In this paper we only applied unweighted ensembles and a heuristic, rather inflexible weighting scheme based directly on past average performance. More sophisticated weighting schemes have been explored in Taylor and Taylor (2021) and Ray et al (2022) using data from the US COVID-19 Forecast Hub. Their results indicate that when some contributing forecasters have a stable record of good performance, giving these more weights can result in improved performance. In particular, restricting the ensemble to a set of well-performing models may be beneficial, a strategy employed in

the *relative WIS weighted median ensemble* (Ray et al 2022) used by the US COVID-19 Forecast Hub since November 2021.

During our study period, only a manual plausibility screening was performed to eliminate highly implausible forecasts. We added a sentence to highlight this choice:

In the study protocol, the median ensemble was defined as our primary ensemble approach as it can be assumed to be more robust to occasional misguided forecasts (e.g., due to technical errors).

Concerning the importance of different models in the ensemble we now provide a supplementary figure with the weights used in the inverse-score approach `KITmetricslab-inverse_WIS_ensemble` in different weeks (Supplementary Figures S9 and S10). It can be seen that weights fluctuate considerably, and no clear patterns emerge. This, of course, is somewhat disappointing as it implies that meaningfully improving performance by weighting the different models in a data-driven way is challenging. We added the following paragraph in the results section to cover this aspect:

The `KITCOVIDhub-inverse_wis_ensemble`, which is an attempt to weigh member models based on recent performance, does not yield any clear benefits over the unweighted median and mean ensembles. As can be seen from Supplementary Figures S9 and S10, the weights fluctuate substantially, implying that the relative performance of different models may be too variable for performance-based weights to pay off.

3: *It is in general difficult to predict the peak and, after that, the evolution of the epidemic. Some of the considered models require some time to adjust if some conditions change. I am wondering if the uncertainty surrounding the estimates at the peak, inflation points or when a new variant arises, is larger than "usual". This is because even looking at the uncertainty surrounding the estimates may be an indication, an alert of something changing.*

Reply: We agree that this is an interesting point to explore. We are currently collaborating with the team of the US COVID-19 Forecast Hub to examine this on a considerably larger archive of forecasts than our German and Polish project can provide. For the present study we have performed some exploratory analyses, see reply to Reviewer 1, remark 2. It does not seem like there is a clear relationship between the disagreement between different models and the possibility of a trend change, but we hope that the planned larger study will shed more light on this question. We have included this aspect as an open question into the discussion section:

An interesting question for future work is whether turning points are preceded by stronger disagreement between models, in which case this might serve as an alert; or whether, on the contrary, trend changes are followed by increased disagreement. Especially the latter question has received considerable attention in economic forecasting (Coibion et al 2012).

4: *At last, you discuss short-term forecasts, with a focus on 1 and 2 weeks ahead forecasts. Does these forecasts depend on the length of training data considered? Could you define an optimal length or is it always better to consider the entire data from a wave?*

Reply: The choice of training data is indeed an important one for the contributing modelling teams. Most of them actually do not fit their models to the entire time series, but just to a recent subset or using sliding windows to let parameters vary over time. This is an important difference between the prediction of emerging diseases such as COVID-19 and seasonal diseases like influenza, where much longer historical time series can be used. Most teams use heuristic strategies in order to choose the length of training data, often manually defining time windows over which parameters can be assumed to be constant (this is the case e.g., for the `Karlen-pypm` model).

We now mention this difficulty in the context of time varying reporting rates (discussion section), which in our opinion are among the key reasons to discard older data:

Another difficulty of case forecasts is the incompleteness ascertainment, which must be assumed to vary over time (see e.g., the discussion in Arik et al 2021). As a consequence, data can be difficult to compare across different phases of the pandemic, and modellers often choose to only use a recent subset of the available data to calibrate their models.

Further changes

As in the time since first submission several publications and preprints on the included submitted models have appeared we have added references to Table 3.

REVIEWERS' COMMENTS:

Reviewer #1 (Remarks to the Author):

The authors have adequately addressed my comments, whether it was adding text to the manuscript, updating figures, or addressing in the rebuttal letter. I very much appreciate the authors' efforts, and think the result manuscript is very nice.

Reviewer #2 (Remarks to the Author):

Overall, I am happy with the responses to my questions and concerns, and I suggest acceptance.

Reviewer #3 (Remarks to the Author):

The authors have satisfactorily addressed my comments.

Reviewer #4 (Remarks to the Author):

I am happy with the current version of the manuscript. All the comments I raised have been properly addressed.

COMMSMED-21-0564-T: Point-by-point response to the referees' comments

Johannes Bracher, Daniel Wolfram, Jannik Deuschel,
Konstantin Görden, Jakob Ketterer et al.

25th August 2022

We would like to thank the review team for the constructive reports throughout the process.

Reviewer 1: The authors have adequately addressed my comments, whether it was adding text to the manuscript, updating figures, or addressing in the rebuttal letter. I very much appreciate the authors' efforts, and think the result manuscript is very nice.

Reviewer 2: Overall, I am happy with the responses to my questions and concerns, and I suggest acceptance.

Reviewer 3: The authors have satisfactorily addressed my comments.

Reviewer 4: I am happy with the current version of the manuscript. All the comments I raised have been properly addressed.